# Is the near-spherical shape the "new black" for smoke?

Anna Gialitaki[1,2], Alexandra Tsekeri[1], Vassilis Amiridis[1], Romain Ceolato[3], Lucas Paulien[3], Anna Kampouri[1,2], Antonis Gkikas[1], Stavros Solomos[4,1], Eleni Marinou[5,1], Moritz Haarig[6], Holger Baars[6], Albert Ansmann[6], Tatyana Lapyonok[7], Anton Lopatin[8], Oleg Dubovik[7], Silke Groß[5], Martin Wirth[5] and Dimitris Balis[2]

[1] National Observatory of Athens / IAASARS, Athens, Greece
[2] Laboratory of Atmospheric Physics, Physics Department, Aristotle University of Thessaloniki, Greece
[3] ONERA, The French Aerospace Lab, Toulouse, France
[4] Research Centre for Atmospheric Physics and Climatology, Academy of Athens, Athens, Greece
[5] Institute of Atmospheric Physics, German Aerospace Center (DLR), Oberpfaffenhofen, Germany
[6] Leibniz Institute for Tropospheric Research (TROPOS), Leipzig, Germany
[7] Laboratoire d'Optique Atmosphérique, CNRS/Université Lille, Villeneuve d'Ascq, France
[8] GRASP-SAS, Villeneuve d'Ascq, France

*Correspondence to:* Anna Gialitaki (togialitaki@noa.gr)

**Abstract.** We examine the capability of near-spherical-shaped particles to reproduce the triple wavelength Particle Linear Depolarization Ratio (PLDR) and Lidar Ratio (LR) values measured over Europe for stratospheric smoke originating from Canadian wildfires. The smoke layers were detected both in the troposphere and the stratosphere, though in the latter case the particles presented PLDR values of almost 18% at 532 nm as well as a strong spectral dependence from the UV to the Near-IR. Although recent simulation studies of rather complicated smoke particle morphologies have shown that heavily coated smoke aggregates can produce large PLDR, herein we propose a much simpler model of compact near-spherical smoke particles. This assumption allows for the reproduction of the observed intensive optical properties of stratospheric smoke, as well as their spectral dependence. We further examine whether an extension of the current AERONET scattering model to include the near-spherical shapes, could be of benefit to the AERONET retrieval for stratospheric smoke cases associated with enhanced PLDR. Results of our study illustrate the fact that triple wavelength PLDR and LR lidar measurements can provide us with additional insight when it comes to particle characterization.

## 1 Introduction

Particles originating from biomass burning activities are known to have a significant effect on radiation and climate (Kaufman et al., 2002). The factors affecting the optical properties of smoke are mainly the black carbon fraction and the impact of the ageing processes (Amiridis et al., 2009). Various findings from field measurements suggest that the smoke particles' surface may serve as highly effective cloud nuclei (Ackerman, 2000; Hoose and Möhler, 2012; Koch and Del Genio, 2010; Marinou

et al., 2019; Nichman et al., 2019), modifying cloud properties and lifetime and thus indirectly affecting the radiative budget. Their various impacts depend also on their lifetime, since they tend to alternate their properties i.e. become less absorbing or more hydrophilic due to atmospheric processes (Amiridis et al., 2009; Adachi and Buseck, 2011).

Smoke particles in the atmosphere can be identified with lidar measurements which provide valuable information on the optical properties of aerosols, such as the depolarization of the backscattered light in terms of the Particle Linear Depolarization Ratio (PLDR). Spherical particles do not depolarize the incident radiation, hence the PLDR can be used to derive information on morphologically complex particles such as smoke. Fresh smoke tends to form fluffy, mostly hydrophobic aggregates composed of many single small monomers. As the particles age in the atmosphere, this aggregate structure collapses, the particles become more hydrophilic and are frequently found covered by cells composed of water soluble components such as sulphates or organic materials (Worringen et al., 2008; Wu et al., 2016). Owning to the aforementioned processes, the PLDR of smoke particles may present a large variability related to the age of the particles (Baars et al., 2019), the presence of other aerosol types found inside the smoke layers (Tesche et al., 2009; Groß et al., 2011) or even the particle water uptake due to different humidity conditions (Cheng et al., 2014). These processes alter smoke particle shape, size and composition, resulting in PLDR values that may vary from 2 to 10% at 532 nm for aged and fresh smoke. These values can be even lower/higher in cases of mixtures with low/high depolarizing components, respectively (i.e. marine/dust particles). Müller et al. (2005) carried out an extensive study on the optical properties and the effect of atmospheric ageing of long-range-transported smoke from Siberia and Canada and found that PLDR at 532 nm did not exceed 1–3 % for 10-day-old plumes. This is comparable to findings by Nicolae et al. (2013), showing that smoke plumes up to 4-day-old present PLDR values of almost 4% at 532 nm. Moreover, measurements conducted in South Africa (Giannakaki et al., 2016) showed that for pure smoke the PLDR values at 355 nm are less than 6%. On the other hand, smoke PLDR has been found to reach values up to 12–14 % at 532 nm if significant concentrations of highly depolarizing components (i.e. soil or dust particles) exist inside lofted smoke layers (Tesche et al., 2009; Veselovskii et al., 2016).

Lately, there have been observational evidence of smoke originating from large-scale fires with PLDR values that exceed the typical range. For example, in Sugimoto et al. (2010) values of 12–15 % at 532 nm are presented for both tropospheric and stratospheric smoke plumes reaching from Mongolia to Nagasaki and Tsukuba in 2007. Nisantzi et al. (2014) reported values of 9–18 % at 532 nm for smoke originating from Turkish fires and observed above Cyprus after 1 to 4 days of transport. A spectral dependence of smoke PLDR with decreasing values from UV to Near-IR was presented for the first time by Burton et al. (2015). The measurements were performed above Denver, Colorado with an airborne HSRL instrument during the DISCOVER-AQ (Deriving Information on Surface Conditions from Column and Vertically Resolved Observations Relevant to Air Quality) field mission. This particular smoke plume was found at 8 km height, originating from Pacific Northwest wildfires and exhibited PLDR values of 20%, 9.3% and 1.8 % at 355, 532 and 1064 nm, respectively.

In the past, many studies have used simpler or more complicated particle shape models in order to reproduce the lidar measurements of smoke. In Kahnert (2017), the PLDR of black carbon aggregates covered by a coating of sulphates was simulated by two different models; a closed cell model (i.e. each monomer in the aggregate is coated separately) and a coated aggregate model (i.e. the whole aggregate is coated). Their analysis showed that for thicker coating the coated cell model of

volume equivalent radius of 0.3 to 0.4 μm, can provide PLDR values of the order of 15% at 532 nm. Mishchenko et al. (2016) and Liu and Mishchenko (2018) used rather complex morphologies for smoke particles, in order to reproduce the PLDR values measured by Burton et al. (2015). Amongst others, these morphologies included a) a fractal aggregate partially embedded in a spherical sulphate cell, b) two-externally-mixed spherical sulphate cells, each hosting an aggregate (models 6 and 11 in Fig. 1 in Liu and Mishchenko (2018) and c) a high-density aspherical soot core, encapsulated in a circumscribing spheroid cell (with

axial ratio of 0.9 to 1.2; model 4 in Fig. 2 in Mishchenko et al., 2016). All these morphologies reproduced successfully the smoke optical properties measured by Burton et al. (2015). Moreover, Luo et al. (2018) used twenty different configurations of coated fractal aggregates and showed that for relatively small fractal dimension (i.e. relatively fresh aggregates), and for small black carbon fractions (i.e. densely coated aggregates; configuration C in Fig. 2 in Luo et al., 2018), the PLDR values can reach up to 40, 15 and 6% at 355, 532 and 1064 nm, respectively. Ishimoto et al. (2019) used fractal aggregates and

artificial surface tension induced on the particles to mimic the effect of coating by water soluble materials forming around the particles. This study present results for both the PLDR and the Lidar Ratio (LR), which is indicative of the composition of the particles. In Liu and Mishchenko (2019), tar ball aggregates were used to model exceptionally strong PLDR as those measured by Burton et al. (2015). The aforementioned studies highlighted the fact that in order to reproduce significant PLDR values (higher than 20% at 532 nm), the fractals need to be coated (i.e. shapes of "Type-B, size 11, Vr = 20" shown in Fig. 4 of

Ishimoto et al., 2019). We should point out though that most of the aforementioned studies refer to monodispersed particles, and averaging over size could possibly supress some of the observed features.

In the spotlight of the large scale Canadian fires of 2017, the discussion regarding the high PLDR values and their spectral dependence for smoke has been opened also for stratospheric smoke. These wildfires inserted large amounts of smoke to the lower stratosphere by explosive Pyro-cumulonimbus activity (Khaykin et al., 2018). In fact, the smoke load in the stratosphere

was found to be comparable to that of a moderate volcanic eruption (Peterson et al., 2018). The smoke plumes encircled the Northern hemisphere in nearly 20 days, reaching Europe in less than 10 days. Above Europe, their properties were intensively studied by the European Aerosol Research Lidar Network (EARLINET; Pappalardo et al., 2014). Multi-wavelength lidar measurements in Central (Ansmann et al., 2018; Haarig et al., 2018; Hu et al., 2019) and South Europe (Sicard et al., 2019) revealed high PLDR values at 355 and 532 nm and a strong spectral dependence from the UV to the Near-IR. However, despite

of the extensive analysis of this event, the microphysical characterization of the stratospheric smoke particles is not yet adequate and further analysis is imperative to draw conclusions. Most of the microphysical properties reported for the stratosphere are retrieved from lidar measurements using inversion algorithms and assumed scattering models that are applied

in EARLINET (e.g. Dubovik et al., 2006; Veselovskii et al., 2002). For example, the derived microphysical properties presented in Haarig et al. (2018) and Hu et al. (2019) are based on the lidar backscatter and extinction coefficient profiles that were used as inputs to inversion schemes. However, the observed PLDR values could not be reproduced by these studies due to the assumed shapes.

In contrast to prior studies, for our investigation for the stratospheric smoke originating from the Canadian wildfires, we do not adopt morphologically complex shapes of bare or coated smoke aggregates, which are associated with excessive computations. Instead, we propose a much simpler model of compact near-spherical particles. Our starting point and main assumption is that the particle near-spherical-shape can be highly depolarizing, as shown in the work of Mishchenko and Hovenier (1995) and Bi et al. (2018). Our analysis shows that for the Canadian stratospheric smoke observed above Europe in

August 2017, the PLDR and LR measurements along with their spectral dependence, can be successfully reproduced with the proposed model of compact near-spherical particles. The size and refractive index of the particles are estimated as well, and seem to agree well with past observations for aged smoke. We further examine the capability of this model to be used on an operational level and in particular as an extension to the AERONET operational aerosol retrieval (Dubovik et al., 2006), since it provides a much simpler and faster solution with respect to more complicated shapes for stratospheric smoke particles (e.g.

Mishchenko et al., 2016; Ishimoto et al., 2019).

Our paper is organized as follows: in Sect. 2 we discuss the methodology followed for the retrieval of the microphysical properties of stratospheric smoke, by constructing look-up-tables of PLDR and LR at 355, 532 and 1064 travnm, assuming (a) near-spherical shapes, and (b) more complicated Chebyshev particle shapes. In Sect. 3 we provide a brief description of the Canadian wildfires during August 2017, describing the mechanism that introduced the smoke particles into the lower

stratosphere and the route of the smoke plume from Canada to Europe. The lidar measurements performed over Leipzig, Germany are presented in this Section. In Sect. 4 we provide the results of our microphysical retrieval. The discussion of these results and the future perspectives of our work are found in Sect. 5. Conclusions are summarized in Sect. 6.

## 2 Construction of look-up-tables

For the retrieval of the smoke microphysical properties from the measured PLDR and LR at 355, 532 and 1064 nm, we

constructed appropriate look-up-tables using near-spherical shapes and more complicated shapes (i.e. Chebyshev particles), along with a range of size distributions and refractive indices based on values reported in the literature for smoke particles (Dubovik et al., 2002; Müller, 2005; Müller et al., 2007a; Nicolae et al., 2013; Giannakaki et al., 2016). For the construction of the look-up-tables we used the T-matrix code (Mackowski and Mishchenko, 1996; Mishchenko and Travis, 1998). The T-matrix outputs are used to calculate PLDR and LR as shown in Eq. (1) and (2):

$$PLDR\ (\lambda) = \frac{P_{11}(180°) - P_{22}(180°)}{P_{11}(180°) + P_{22}(180°)} \tag{1}$$

$$LR\ (\lambda) = \frac{4\pi\ C_{ext}(\lambda)}{C_{sca}(\lambda)\ P_{11}(180°)} \tag{2}$$

where $P_{ij}$ are the elements of the scattering matrix, $C_{ext}$ and $C_{sca}$ are the extinction and scattering cross sections, and $\lambda$ is the wavelength (Fig. 1).

### 2.1 Near - spherical shapes

5  We modelled the near-spherical shapes using spheroid particles with different axial ratios $\varepsilon$. The axial ratio of a spheroid is defined as the ratio of the ellipse rotational axis ($a$) to the axis perpendicular to the rotational axis ($b$) as $\varepsilon = {}^{a}/_{b}$. If $\varepsilon > 1$ then the spheroid is characterized as prolate, whereas if $\varepsilon < 1$, the spheroid is characterized as oblate (Mishchenko et al., 2002; Dubovik et al., 2006). To describe the spheroidal shape in the spherical coordinate system we use Eq. (3) where $r$ is the radius of the volume equivalent sphere and $\theta, \varphi$ are the zenith and azimuth angles respectively.

$$r\ (\theta,\varphi) = a\ \left[ \sin^2\theta + \frac{a^2}{b^2}\ \cos^2\theta \right]^{-1/2} \tag{3}$$

For the present study we used $\varepsilon$ values from 0.6 to 1.55. Figure 2 presents some examples of the near-spherical shapes used, embedded in a perfectly spherical shell to demonstrate their deviation from the perfect sphere.

We assumed that the shape distribution of the near-spherical particles is a mono-modal, normal distribution $n_s(\varepsilon)$ as shown in

15  Eq. (4), with $\sigma_s$ the sigma of the distribution fixed to 0.05, and $\varepsilon_s$ the mean axial ratio (Table 1). We also assume that the shape distribution does not change with particle size. The fixed width of the shape distribution $\sigma_s$ is necessary for the reduction of the retrieval complexity. Its' small value is used to avoid the wash-out of the characteristic optical properties which are shown for a relatively narrow axial ratio range for near-spherical particles (e.g. Bi et al., 2018).

$$n_s\ (\varepsilon) = \frac{1}{\sqrt{2\pi}\ \sigma_s} \exp\left( -\frac{(\varepsilon - \varepsilon_s)}{2\sigma_s^2} \right) \tag{4}$$

The size distributions considered for the near-spherical particles are mono-modal and log-normal with mean geometric radius $r_g$ and geometric standard deviation $\sigma_g$, as shown in Eq. (5). The grid used for $r_g$ is 0.1–0.7 μm, while $\sigma_g$ is fixed at 0.4. The fixed width of the size distribution $\sigma_g$ is again a simplification we used in order to reduce the retrieval complexity, considering

that this parameter does not greatly affect the lidar-derived optical properties (e.g. Burton et al., 2016). Choosing a log-normal size distribution over any other plausible type of distribution is not expected to alter our results significantly (Hansen and Travis, 1974).

$$n\,(r) = \frac{1}{\sqrt{2\pi}\,r\,\sigma_g}\exp\left[-\frac{1}{2}\left(\frac{\ln\,(r/r_g)}{\sigma_g}\right)^2\right] \tag{5}$$

Moreover, a wavelength-independent complex refractive index $m$ was assumed, with real part ($mrr$) varying from 1.35 to 1.85 and imaginary part ($mri$) varying from 0.005 to 0.5 (Dubovik et al., 2002; Müller et al., 2005; Nicolae et al., 2013; Giannakaki et al., 2016). An overview of the values used for the generation of the look-up-tables for the near-spherical particles is presented in Table 1.

## 2.2 Chebyshev particles

In order to investigate whether particles of more complicated shapes than the near-spherical shape can reproduce both the PLDR and LR measurements of stratospheric smoke, we also constructed look-up-tables for smoke particles resembling "Chebyshev particles" using the T-matrix code. Chebyshev particles (Fig. 3) are produced by the deformation of a sphere by means of a Chebyshev polynomial. In the spherical coordinates system, their shape is described as shown in Eq. (6), where $r_0$ is the radius of the perfect sphere, $u$ is the deformation parameter and $T_n(cos\theta)$ is the Chebyshev polynomial of degree $n$

(Mishchenko and Travis, 1998).

$$r\,(\theta,\varphi) = r_0\,(1 + u\,T_n\,(cos\theta)), \quad |u|<1 \tag{6}$$

Only Chebyshev polynomials of second ($T_2$) and fourth ($T_4$) degree were used, with deformation parameter values of $u = \pm$ 0.05, $\pm$ 0.10, $\pm$ 0.15, $\pm$ 0.20, $\pm$ 0.25 and $u = \pm$ 0.05, $\pm$ 0.10, $\pm$ 0.15 respectively. We considered the same refractive indices as the ones used for the generation of the look-up-tables of the near-spherical particles, while for the size distribution we used also mono-modal, log-normal distributions. Table 1 summarizes the properties used for the construction of the look-up-tables

for Chebyshev particles.

## 3 Description of the dispersion and vertical distribution of smoke

The extreme pyro-convection (Fromm et al., 2010) that was recorded in the area of British Columbia (western Canada) during summer 2017, resulted in particularly strong updrafts that penetrated and released large amounts of smoke particles into the lower stratosphere (Peterson et al., 2018). Here we use an ensemble of satellite observations from MODIS (Moderate

Resolution Imaging Spectroradiometer) on board Terra and Aqua, OMPS (Ozone Mapping and Profiler Suite) on board Suomi NPP and CALIOP (Cloud-Aerosol Lidar with Orthogonal Polarization) on board CALIPSO, to identify the dispersion and vertical distribution of the plume above Canada. The combination of these observations is shown in Fig. 4, where true-color images from MODIS are overlaid with the fire active regions and thermal anomalies (red dots) from Suomi NPP, and CALIPSO (green lines) overpasses on 8 and 15 August 2017.

Figures 5 and 6 show the backscatter coefficient and PLDR curtain plots at 532 nm from CALIPSO measurements. Based on these observations smoke plumes were found above the regions of fire activity since the beginning of August (Fig. 4a), when the plumes remained in the troposphere, below 5–6 km (39º–45º N, 123º–125º W) (Fig. 5a, red dashed lines), exhibiting low PLDR values of the order of 3–4 % (Fig. 5b) at 532 nm. Then on 12 August 2017, the unprecedented buoyancy force caused by the strong fire activity started lifting the plumes up towards the tropopause, while already on 15 August 2017 smoke covered a large part of North Canada (Fig. 4b). CALIPSO observations on 15 August reveal that the plume lies into the stratosphere at 11–14 km height (63º–69º N, 89º–94º W) (Fig. 6a, red dashed lines) and PLDR values exceed 15% at 532 nm (Fig. 6b).

Owning to the altitude of the smoke plume, one could attribute such PLDR values to the beginning of ice formation. Indeed, radiosonde temperature profiles from two stations located underneath the smoke plume (green stars in Fig.4b), reveal that the temperature above 11 km drops below - 40°C, at which point homogeneous ice formation can occur (Wallace and Hobbs, 2006). However, the PLDR values of cirrus clouds are usually no less than 40% at 532 nm (Chen et al., 2002; Noel et al., 2002; Voudouri et al., 2020) whereas the values observed in this case are mostly between 15 and 25% at 532 nm, and remain so during the months of August and September following the stratospheric injection (Baars et al., 2019; Hu et al., 2019). Further analysis of CALIOP data provides a mean (median) value of the backscatter related Angstrom exponent (BAE) at 532/1064 nm of 0.9 (0.9) with a standard deviation on 1.07. For cirrus clouds, BAE values close to zero are expected, although, as indicated by the large standard deviation, CALIPSO data are highly noisy at these altitudes. A recent study by Yu et al. (2019) also showed that the largest fraction of stratospheric smoke particles consisted of organic carbon (98% compared to 2% for black carbon). Particles of such high organic carbon content serve poorly as ice nuclei (Kanji et al., 2017; Phillips et al., 2013). Although the possibility of small ice crystals formed inside the smoke layers cannot be excluded, (largely due to the absence of in situ measurements) the aforementioned characteristics indicate that this plume consists primarily of smoke particles rather than ice crystals.

Inside the lower stratosphere, unaffected by the intensive tropospheric interactions, smoke particles started drifting, following a North-Easterly direction and first appeared over Europe approximately after mid-August (Khaykin et al., 2018; Ansmann et al., 2018; Hu et al., 2019).

Interestingly, even after two months of the initial stratospheric smoke injection the plume seems to have sustained its high depolarization capability. During this period the smoke plume has already encircled the Northern hemisphere and it was detected by airborne lidar measurements performed above the Atlantic near the west coast of Ireland (Fig. 7b). Lidar

observations showed PLDR values in the range of 10–14 % at 532 nm between 10 and 12 km (Fig. 7a) These observations were conducted in the framework of Wave-driven ISentropic Exchange (WISE) mission organised by the German Aerospace Centre (DLR) and support the high depolarization values detected for months over Europe by EARLINET, as shown in Fig. 7 in Baars et al. (2019).

## 3.1 Lidar measurements in Leipzig

The highest smoke load over EARLINET was been reported at Leipzig, Germany (Ansmann et al., 2018a; Baars et al., 2019). Measurements at the Leibniz Institute of Tropospheric Research (TROPOS) were performed with the BERTHA (Backscatter Extinction lidar-Ratio Temperature Humidity profiling Apparatus) multi-wavelength polarization Raman lidar system. The system measures the total and cross-polarized component of the elastic backscattered light at 355, 532 and 1064 nm, which are used to derive the PLDR at these wavelengths. It is also able to perform independent measurements of the aerosol extinction coefficient at 387, 607 nm and (after optics re-arrangement) at 1058 nm, and thus has the capability to provide the LR profiles at 355, 532 and 1064 nm (Haarig et al., 2017). On 22 August 2017, the profiles of the stratospheric smoke backscatter and extinction coefficients at 355, 532 and 1064 nm and the smoke PLDR at 355 and 532 nm were derived from two-and-a-half-hour averaging of the lidar signals between 20:45 and 23:17 UTC. The PLDR value at 1064 nm was calculated using a forty-minute averaging between 23:50 and 00:30 UTC (Haarig et al., 2018). The gap between the end of the first measurement and the beginning of the second, corresponds to the necessary time for the rearrangement of BERTHA optics. To ensure the high quality of depolarization measurements, the $\Delta\pm45$ depolarization calibration method proposed by Freudenthaler et al., (2009) was followed, while the effect of different parameters on the depolarization measurements of the BERTHA lidar system has been carefully assessed and is presented in detail in Haarig et al. (2017).

Layer-integrated values of PLDR and LR for the stratospheric smoke layer are shown in Fig. 8 and Table 2 along with their associated uncertainties. The derived LRs are typical for aged Canadian smoke at 355 nm (40 ± 16 sr) and 532 nm (66 ± 12 sr) (Müller et al., 2005; 2007b). Low signal-to-noise ratio at the plume height prevented detailed retrievals of particle extinction coefficient at 1058 nm. Thus, for the LR values at 1064 nm only few measurement points could be derived (Haarig et al., 2018). This yields a LR value of 92 ± 27 sr at 1064 nm. The increasing tendency of the LR from the UV to the visible part of the spectrum has been also reported before for aged Canadian smoke (Müller et al., 2005; 2007b). Measurements reported in Haarig et al. (2018) suggest that there is an increase also at the Near-IR, although there are currently no other available measurements of the LR of smoke particles at this wavelength. On the other hand, the PLDR values of stratospheric smoke are much larger than those usually reported in the past for tropospheric smoke. The layer-integrated PLDR value at 355 nm is 22.4 ± 2.5 %, decreasing to 18.4 ± 1.2 % at 532 nm and 4 ± 2.3 % at 1064 nm. The uncertainties in PLDR values include both the systematic errors and the standard deviation of the measurements.

These results are in agreement with the PLDR values measured above Lille and Palaiseu for the period 24 to 31 August, 2017 (Hu et al., 2019). To the best of our knowledge, up to now the majority of observations of such smoke PLDR values, refer to smoke particles found in the stratosphere (i.e. Ohneiser et al., 2020). The sole exception is the case study reported by Burton et al. (2015) (see also Table 2).

## 4 Smoke microphysical retrieval

### 4.1 Near-spherical particles

First, we present the smoke microphysical retrieval considering the near-spherical shape for the smoke particles, as described in section 2.2. All the possible solutions are selected from the pre-calculated T-matrix look-up-tables, based on Eq. (7). For each measured PLDR and LR, at each wavelength $\lambda$, the simulated value must be within the corresponding measurement error $e$.

$$\left| \delta_\lambda^M - \delta_\lambda^S \right| \leq e(\delta_\lambda^M) \text{ and } \left| LR_\lambda^M - LR_\lambda^S \right| \leq e(LR_\lambda^M) \tag{7}$$

Where $M$ denotes to measured PLDR and LR at wavelength $\lambda$ = 355, 532 and 1064 nm and $S$ denotes to the corresponding simulations. The solution is selected amongst the possible solutions based on the minimization criteria of Eq. (8) (see also Fig. 1).

$$\sum_{\lambda = 355, 532, 1064} \left( \left( \frac{\delta_\lambda^M - \delta_\lambda^S}{e(\delta_\lambda^M)} \right)^2 + \left( \frac{LR_\lambda^M - LR_\lambda^S}{e(LR_\lambda^M)} \right)^2 \right) = \mathbf{min} \tag{8}$$

Following this methodology, for the near-spherical particles ten possible solutions were found to reproduce the measurements within the measurement uncertainty. These are listed in Table 3 along with the resulting cost functions calculated with Eq. (8). For these solutions, the mean axial ratio $\varepsilon_s$ of the particles covers the range 1.1 to 1.4 while the range of the mean geometric radius $r_g$ is 0.25 μm (respective effective radius: $reff$ = 0.4 μm) up to 0.45 μm ($reff$ = 0.7 μm). For the complex refractive index $m$, the imaginary part $mri$ does not exceed the value of i0.03, while the real part $mrr$ takes values from 1.35 to 1.55. The minimization of the cost function (Eq. 8) is achieved for near-spherical particles with $\varepsilon_s$ = 1.4, $m$ = 1.55 + i0.025 and $r_g$ = 0.25 μm, suggesting a strong accumulation mode for the size distribution of the particles, with sufficiently small $mri$ so as the characteristic enhancement in PLDR does not wash out due to the strong absorption (Bi et al., 2018). All possible solutions as well as the solution that minimizes the cost function are presented in Fig. 9 and 10.

### 4.2 Chebyshev particles

For Chebyshev particles of second ($T_2$) and fourth degree ($T_4$) used herein, the search in the constructed look-up-tables provided the solutions listed in Table 4. For all the solutions, deformation parameter for Chebyshev particles of the second degree ranges from $u = -0.25$ to $0.15$, while for particles of the fourth degree only one solution was found with $u = -0.1$. These $u$ values suggest small deviations from sphericity, meaning that these morphologies also resemble near-spherical shapes (see also Fig. 3). Only for two cases the size of the particles was found to be larger than the size of the near-spherical shaped particles. In particular the range of $r_g$ was from 0.15 µm ($reff = 0.2$ µm) to 0.55 µm ($reff = 0.8$ µm). The complex refractive index, in some cases exceed the corresponding values for near-spherical particles. The range of the imaginary part $mri$ is from 0.005 to 0.055, and the range of the real part $mrr$ is from 1.35 to 1.8. The minimization of the cost function (Eq. 8) is achieved for Chebyshev particles of the second degree with u = - 0.25 (resembling an oblate near-spherical particle), complex refractive index $m = 1.65 + i0.03$ and mean geometric radius $r_g = 0.2$ µm (Fig. 11). For Chebyshev particles of the fourth degree, the sole solution presented values of $u = -0.1$, $m = 1.35 + i(0.01)$ and $r_g = 0.55$ µm (Fig. 12).

### 4.3 More case studies

Although the available literature on the PLDR and LR values of stratospheric smoke is for now limited, we see that we can reproduce all reported PLDR and LR listed in Table 2, using the near-spherical shape model (Fig. S4–S9 in the Supplement). All cases listed in Table 2 are associated with Pyro-cumulonimbus activity. As already mentioned the case studies of Burton et al. (2015), Hu et al. (2019) and Haarig et al. (2018) refer to Canadian smoke, while the most recent case study presented by Ohneiser et al. (2020) refer to the Australian wildfires of 2019-2020. Tables S4–S9 in the Supplement present the properties of near-spherical particles and Chebyshev particles that reproduce the PLDR and LR observations reported in the aforementioned studies. Results are in line with the results presented for Haarig et al. (2017).

We note here that all the retrievals indicate fine particles, with mean geometric radius that does not exceed the value of 0.55 µm. The simulations presented by Bi et al., (2018; Fig. 2) suggest that for the near-spherical particles the measured spectral dependence of PLDR (steeply decreasing from the UV to the Near-IR) could not be reproduced by coarse particles. Thus, the possibility of an optically significant coarse mode would have to be investigated with a different shape model. In any case though, the retrieved fine mode is in good agreement with in-situ measurements of aged smoke particles (i.e. Dahlkötter et al., 2014). The presence of a pronounced accumulation mode is also suggested by the extinction related Angstrom exponent (EAE) measured in Leipzig (- 0.3 ± 0.4 at 355/532 nm and 0.85 ± 0.3 at 532/1064 nm). According to Eck et al. (1999), a strong spectral slope in EAE can be associated with a prominent accumulation mode of the size distribution for smoke particles.

## 5 Discussion

### 5.1 Potential of near-spherical model for AERONET products

Up to now, the use of near-spherical particles is found to well-reproduce the lidar measurements of smoke optical properties, as well as their wavelength dependence. In this section, we further extend our study to examine the potential of using the near-spherical shape model with sun-photometer measurements, on an operational level. Our main idea is whether the AERONET non-spherical scattering model could be extended to include also near-spherical particles for stratospheric smoke. In the current AERONET retrieval scheme, non-spherical particles are modelled as spheroids with axial ratios of 0.33 to 0.7 and 1.44 to 2.99, thus omitting the near-spherical particles. These ranges of axial ratios were selected towards an optimized retrieval for dust particles (Dubovik et al., 2006).

As an indication of the limitation of the current AERONET non-spherical model on reproducing the stratospheric smoke cases, we refer to AERONET Version 2 morning observations (05:42 UTC) from Lindenberg site on 23 August 2017 (180 km from Leipzig) and Version 3 noon observations (11:03) from Punta Arenas on 8 January 2020. For these two cases, the sun-photometer measurements should be affected by the presence of stratospheric smoke as shown in Haarig et al. (2018) and Ohneiser et al. (2020). The corresponding AERONET retrievals present residual errors higher than 5%, which marks the threshold of a successful AERONET retrieval (Holben et al., 2006). For the first case over Lindenberg site, the retrievals were rejected from the quality assured Level 2 AERONET products, while they are absent from the latest AERONET Version 3.

The following analysis shows possible benefits for the AERONET retrievals of stratospheric smoke, from including the near-spherical model in the retrieval scheme. Towards this end, we show that the AERONET non-spherical model is limited in reproducing the phase function ($P_{11}$) of particles with near-spherical shapes. We should note here that this is only a first-level approximation of the full solution, since we do not account for the multiple scattering along the column of the sun-photometer measurements, but rather assume only single scattering.

In the following we tried to reproduce the $P_{11}$ of the near-spherical stratospheric smoke particles presented herein, using the $P_{11}$ calculated with the AERONET non-spherical model. For the latter we used the pre-calculated AERONET Kernels (Dubovik et al., 2006), for a large suite of refractive indices and size distributions (Table 6). The comparison is performed for the sun-photometer wavelengths 440, 670, 870 and 1020 nm. Figure 13 (left plot) shows the $P_{11}$ at 440 nm calculated for the near-spherical stratospheric smoke particles (purple line in the plots), and the comparison with the $P_{11}$ at 440 nm calculated using the AERONET non-spherical model (blue lines) with $r_g = 0.25$ μm and all refractive indices listed in Table 6. The complete set of calculations (for all $r_g$ and refractive indices listed in Table 6, and for AERONET wavelengths of 670, 870 and 1020 nm) is provided in the Supplement (Fig. S14–S69). Figure 13 shows also the degree of linear polarization (-$P_{12}/P_{11}$)

(middle plot) and the values of $P_{22}/P_{11}$ (right plot). These plots are provided to show the potential of polarized measurements in better discerning the features of near-spherical particles (as is the case with the PLDR measurements).

In order to quantify the residual ($Err$) of the fitting we use Eq. 9 (https://aeronet.gsfc.nasa.gov).

$$Err = \sqrt{\frac{\sum_{i=1}^{n}(lnf^* - lnf)^2}{N}} * 100 = \%Error \qquad (9)$$

Where $lnf^*$ denotes to $P_{11}$ values calculated with the near-spherical model, $lnf$ denotes the $P_{11}$ values calculated with the AERONET non-spherical model, and $N$ is the number of values, in terms of wavelengths and scattering angles. $Err$ is calculated considering the four AERONET wavelengths at 440, 670, 870 and 1020 nm and the scattering angles from 0° to 150°, which indicate the measurement geometry of the AERONET sun-photometers.

The residuals for fitting the phase function of the near-spherical particles with the AERONET non-spherical model, are presented in Fig. 11. The minimum $Err$ is 9.4%, whereas, the limit of a successful AERONET retrieval is 5% (Holben et al., 2006), indicating the limitations of the AERONET non-spherical model in reproducing the phase function of near-spherical smoke particles. Similar results for the $Err$ considering only the wavelengths at 440, 670, 870 and 1020 nm are provided in the Supplement (Fig. S10–S13).

Again, we should emphasize the fact that the residual threshold of 5% denotes to the multiple-scattered light, which may mask the differences seen in the single-scattering properties in Fig. 12 and 13. In order to have a clear understanding of whether the near-spherical shape model could in fact improve the AERONET retrieval for stratospheric smoke, further analysis is imperative. For example, although a large range of the parameters affecting the retrieval and combination of these parameters were used, there are always other possible combinations that were not accounted for. To draw any strong conclusions one

would have to perform a numerical inversion of the stratospheric smoke measurements, and investigate the corresponding residuals. This is part of our future work, continuing the characterization of stratospheric smoke particles with the combination of sun-photometer and lidar measurements.

## 6 Conclusions

The unique optical properties of transported stratospheric smoke, originating from the Pyro-cumulonimbus activity of the large

Canadian fires 2017, were reproduced using T-matrix simulations and assuming near-spherical shapes for smoke. This is consistent with results of past studies showing that near-spherical particles produce PLDR values that can reach up to 100% depending also on their size and composition (Bi et al., 2018) and that smoke particles in particular, when heavily coated or even encapsulated with weakly absorbing materials, can produce large depolarization with a noticeable spectral dependence

(Mishchenko et al. 2016; Ishimoto et al., 2019). As a next step we examined whether the AERONET retrieval could possibly be benefited by taking into account the near-spherical shape for stratospheric smoke. Sun-photometer measurements from Lindenberg and Punta Arenas revealed that for the current algorithm configuration, AERONET retrievals for stratospheric smoke cases are associated with high residual errors (higher than 5%) and are eventually rejected. The extension of the AERONET scattering model to include the near-spherical shapes could possibly improve the retrieval for these cases that seem to become frequent. Our analysis does not mean to generalize on the performance of the AERONET retrieval on tropospheric biomass burning cases. It is focused on the stratospheric smoke cases, related to PyroCb activity.

Concluding, studying the stratospheric smoke from the Canadian wildfire activity provided us with the great opportunity to show the potential of remote sensing measurements in investigating and deducing new optical and microphysical properties for the stratospheric smoke particles. Our analysis highlighted also the need for coordinated ground-based lidar network measurements such as the ones provided by EARLINET, as an exploratory tool in investigating unknown processes in the stratosphere.

*Data availability.* The satellite products used in this study are the CALIPSO 5 km aerosol profile product (Vaughan et al., 2019) publicly available at the AERIS/ICARE database (ICARE data and services center, 2019); the MODIS Corrected Reflectance (True Color) images (Gumley et al., 2010) publicly available on the NASA Worldview center (NASA Worldview snapshots application center, 2019). The HALO-DLR aircraft lidar observations (level 2 data of depolarization and water vapour mixing ratio profiles) used in this study are available via the HALO database (https://halo-db.pa.op.dlr.de/). The AERONET version 3 data are available at https://aeronet.gsfc.nasa.gov/. The pre-calculated AERONET Kernels used in this work are publicly available at https://code.grasp-open.com/open/spheroid-package (last access: 15 July 2020). All datasets created during the calculation of the scattering properties of near-spherical and Chebyshev particles can be accessed through the ReACT-NOA database upon request to the corresponding author.

*Author contributions.* VA, AT and AG conceived the presented idea; VA and AT supported AG on the analysis, manuscript preparation, and figures design; AT guided and supervised AG on the scattering model calculations and results interpretation; RC and LP performed the analysis on fractal aggregates scattering properties and provided AG with the results (not shown in the final manuscript); AG, AK and SS analyzed the MLS water vapour data and performed Flexpart model runs to support the dispersion of the smoke and volcanic plumes (not shown in the final manuscript); EM prepared the CALIPSO data and figures; MH, HB and AA collected and analyzed Leipzig lidar measurements; TL, AL and OD supported AG to confirm T-matrix results for near-spherical particles; SG and MW performed the airborne lidar measurements and the corresponding analysis; DB advised AG on the interpretation of the results of this study. All authors provided critical feedback and helped shape the research, analysis and manuscript.

*Competing interests.* The authors declare that they have no conflict of interest.

*Special issue statement*. This article is part of the special issue "EARLINET aerosol profiling: contributions to atmospheric and climate research". It is not associated with a conference.

*Acknowledgements.* The authors would like to thank M. I. Mishchenko for making the T-matrix codes available (https://www.giss.nasa.gov/staff/mmishchenko/t_matrix.html, last access: 15 July 2020). We are grateful to EARLINET (https://www.earlinet.org/, last access: 15 July 2020), ACTRIS (https://www.actris.eu, last access: 15 July 2020), for the data collection, calibration, processing and dissemination. We are grateful to the AERIS/ICARE Data and Services Center for providing access to the CALIPSO data used and their computational center (http: //www.icare.univ-lille1.fr/, last access: 15 July 2020). We thank the NASA/LaRC/ASDC for making available the CALIPSO products. The authors are grateful to the NASA EOS Aura MLS team for providing free access to the MLS water vapour data (https://mls.jpl.nasa.gov/, last access:

15 July 2020). We acknowledge the use of imagery from the Worldview Snapshots application (https://wvs.earthdata.nasa.gov/, last access: 15 July 2020), part of the Earth Observing System Data and Information System (EOSDIS). We acknowledge the PANhellenic GEophysical observatory of Antikythera (PANGEA) datacenter for supporting all computations for the development of datasets for the calculation of the scattering properties of near-spherical and Chebyshev particles.

*Financial support.* The research leading to these results was supported through the European Research Council (ERC) under the European Community's Horizon 2020 research and innovation framework program – ERC grant agreement 725698 (D-TECT). Anna Gialitaki acknowledges support of this work by the project "PANhellenic infrastructure for Atmospheric Composition and climatE chAnge" (MIS 5021516), which is implemented under the action "Reinforcement of the Research and Innovation Infrastructure", funded by the operational program "Competitiveness, Entrepreneurship and Innovation" (NSRF 2014–2020) and co-financed by Greece and the European Union (European Regional Development Fund). EM was funded by a DLR VO-R young investigator group and the Deutscher Akademischer Austauschdienst (grant no. 57370121). NOA team acknowledges the support of Stavros Niarchos Foundation (SNF).

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

**Table 1.** The parameters used for the generation of the look-up tables of the near-spherical and Chebyshev particles.

| Parameter | Range |
|---|---|
| $r_g$ (μm) [step]; $r_{eff}$ (μm) [step] | 0.1–0.7 [0.05]; 0.15–1.05 [0.07] |
| $\sigma_g$ (fixed) | 0.4 |
| $mrr$ [step] | 1.4–1.75 [0.05] |
| $mri$ [step] | 0.005–0.045 [0.005] and 0.05–0.5 [0.05] |
| $\varepsilon_s$ [step] | 0.6–1.55 [0.05] |
| $\sigma_s$ (fixed) | 0.05 |
| $u$ [step], $T_2$ | $\pm 0.25$ , $\pm 0.20$ , 0.15, $\pm 0.05$ [0.05] |
| $u$ [step], $T_4$ | $\pm 0.25$ , $\pm 0.20$ , 0.15, $\pm 0.05$ [0.05] |

**Table 2.** LR and PLDR layer-integrated mean values at 355, 532 and 1064 nm for the stratospheric smoke layer, on 22 August 2017, at Leipzig, Germany (Haarig et al., 2018). Also shown are the multi-wavelength observations of PLDR and LR reported in previous and later studies for stratospheric or tropospheric smoke particles exhibiting high PLDR values.

| | $PLDR_{355}$ (%) | $PLDR_{532}$ (%) | $PLDR_{1064}$ (%) | $LR_{355}$ (sr) | $LR_{532}$ (sr) | $LR_{1064}$ (sr) |
|---|---|---|---|---|---|---|
| Haarig et al. (2018) | $22.4 \pm 2.5$ | $18.4 \pm 1.2$ | $4 \pm 2.3$ | $40 \pm 16$ | $66 \pm 12$ | $92 \pm 27$ |
| Burton et al. (2015) | $20.3 \pm 3.6$ | $9.3 \pm 1.5$ | $1.8 \pm 0.2$ | | | |
| Hu et al. (2019) | $23 \pm 3$ | $20 \pm 3$ | $5 \pm 1$ | $35 \pm 6$ | $54 \pm 9$ | |
| | $24 \pm 4$ | $18 \pm 3$ | $4 \pm 1$ | $45 \pm 9$ | $56 \pm 12$ | |
| | $28 \pm 8$ | $18 \pm 3$ | $5 \pm 1$ | $34 \pm 12$ | $58 \pm 20$ | |
| Ohneiser et al. (2020) | $23 \pm 4.6$ | $14 \pm 1.4$ | | $83 \pm 24.9$ | $102 \pm 20.4$ | |
| | $20 \pm 4$ | $14 \pm 1.4$ | | $53 \pm 15.9$ | $76 \pm 15.2$ | |
| | $26 \pm 5.2$ | $15 \pm 1.5$ | | $97 \pm 29.1$ | $104 \pm 20.8$ | |

**Table 3.** Calculated properties of near-spherical particles, that reproduce the PLDR and LR at 355, 532 and 1064 nm, as reported in Haarig et al., (2017). Also shown is the corresponding cost function of each solution. The solution that minimizes the cost function (Eq. 8) is highlighted in blue.

| | | | | **Measurements – Leipzig (22 August 2017)** | | | | | | |
|---|---|---|---|---|---|---|---|---|---|---|
| | | | | $PLDR_{355}$ | $PLDR_{532}$ | $PLDR_{1064}$ | $LR_{355}$ | $LR_{532}$ | $LR_{1064}$ | |
| | | | | $22.4 \pm 1.5$ | $41 \pm 16$ | $18.4 \pm 0.6$ | $66 \pm 12$ | $4.3 \pm 0.7$ | $92 \pm 27$ | |
| | | | | **Simulations – Near-spherical particles** | | | | | | |
| $r_g$ | $\varepsilon_s$ | $mri$ | $mrr$ | $PLDR_{355}$ | $PLDR_{532}$ | $PLDR_{1064}$ | $LR_{355}$ | $LR_{532}$ | $LR_{1064}$ | Cost function |
| 0.45 | 1.1 | 0.005 | 1.35 | 23.19 | 17.73 | 2.08 | 33.03 | 67.37 | 118.96 | 2.54 |
| 0.50 | 1.1 | 0.005 | 1.35 | 23.85 | 19.53 | 2.80 | 29.08 | 56.02 | 121.76 | 4.02 |
| 0.35 | 1.2 | 0.020 | 1.45 | 23.21 | 17.22 | 3.89 | 43.14 | 62.77 | 106.10 | 1.48 |
| 0.35 | 1.2 | 0.025 | 1.45 | 23.10 | 17.29 | 3.85 | 54.30 | 75.10 | 117.69 | 3.25 |
| 0.30 | 1.3 | 0.025 | 1.50 | 22.21 | 18.08 | 4.90 | 43.17 | 62.97 | 104.92 | 0.48 |
| 0.30 | 1.3 | 0.030 | 1.50 | 22.35 | 18.31 | 4.87 | 52.55 | 73.40 | 114.38 | 1.74 |
| 0.25 | 1.4 | 0.020 | 1.55 | 21.15 | 17.87 | 4.86 | 33.99 | 55.01 | 90.12 | 1.49 |
| **0.25** | **1.4** | **0.025** | **1.55** | **21.38** | **18.09** | **4.78** | **40.60** | **62.91** | **96.87** | **0.37** |
| 0.25 | 1.4 | 0.030 | 1.55 | 21.61 | 18.31 | 4.70 | 48.15 | 71.64 | 103.84 | 0.81 |

**Table 4.** Calculated properties of Chebyshev particles of second ($T_2$) and fourth ($T_4$) degree, that reproduce the PLDR and LR at 355, 532 and 1064 nm, as reported in Haarig et al., (2017). Also shown is the corresponding cost function of each solution. The solution that minimizes the cost function (Eq. 8) is highlighted in blue.

| | | | | Simulations-Chebyshev particles of 2nd degree | | | | | | |
|---|---|---|---|---|---|---|---|---|---|---|
| $r_g$ | $\varepsilon_s$ | $mri$ | $mrr$ | PLDR$_{355}$ | PLDR$_{532}$ | PLDR$_{1064}$ | LR$_{355}$ | LR$_{532}$ | LR$_{1064}$ | Cost function |
| 0.50 | - 0.05 | 0.015 | 1.4 | 22.59 | 18.05 | 3.30 | 43.95 | 62.86 | 114.13 | 1.08 |
| 0.35 | - 0.10 | 0.020 | 1.45 | 23.94 | 19.03 | 4.31 | 41.38 | 61.94 | 105.71 | 1.04 |
| 0.35 | - 0.10 | 0.025 | 1.45 | 24.18 | 19.10 | 4.27 | 52.32 | 74.01 | 117.19 | 2.76 |
| 0.25 | - 0.20 | 0.030 | 1.60 | 21.47 | 18.59 | 6.42 | 38.73 | 54.84 | 94.68 | 1.90 |
| 0.25 | - 0.20 | 0.035 | 1.60 | 21.44 | 18.86 | 6.35 | 45.44 | 62.15 | 101.45 | 1.43 |
| 0.25 | - 0.20 | 0.040 | 1.60 | 21.44 | 19.11 | 6.26 | 52.96 | 70.14 | 108.40 | 2.37 |
| 0.25 | 0.10 | 0.045 | 1.60 | 22.96 | 17.65 | 4.99 | 45.19 | 58.42 | 106.28 | 1.32 |
| 0.25 | 0.10 | 0.050 | 1.60 | 23.08 | 17.81 | 4.93 | 52.22 | 65.98 | 113.89 | 1.63 |
| 0.20 | - 0.25 | 0.025 | 1.65 | 21.80 | 19.11 | 5.13 | 35.10 | 55.73 | 80.98 | 1.53 |
| **0.20** | **- 0.25** | **0.030** | **1.65** | **21.97** | **19.30** | **5.00** | **40.35** | **61.97** | **85.27** | **0.86** |
| 0.20 | - 0.25 | 0.035 | 1.65 | 22.13 | 19.48 | 4.88 | 46.27 | 68.68 | 89.57 | 1.09 |
| 0.15 | 0.15 | 0.050 | 1.80 | 24.68 | 18.82 | 3.66 | 38.08 | 55.30 | 68.87 | 2.58 |
| 0.15 | 0.15 | 0.055 | 1.80 | 24.87 | 18.94 | 3.59 | 41.63 | 59.64 | 71.03 | 2.16 |
| | | | | Simulations-Chebyshev particles of 4th degree | | | | | | |
| $r_g$ | $\varepsilon_s$ | $mri$ | $mrr$ | PLDR$_{355}$ | PLDR$_{532}$ | PLDR$_{1064}$ | LR$_{355}$ | LR$_{532}$ | LR$_{1064}$ | Cost function |
| 0.55 | - 0.10 | 0.01 | 1.35 | 23.02 | 17.73 | 5.07 | 44.13 | 67.51 | 122.24 | 1.82 |

**Table 5.** Parameters used for the calculations of the optical properties of smoke particles, using the non-spherical model of AERONET, in Fig. 12 and 13.

| | |
|---|---|
| $r_g (\mu m)$ | 0.1, 0.15, 0.2, 0.25, 0.3, 0.4, 0.5, 0.8, 1.0, 1.5, 2.0, 2.5, 3.0, 4.0 |
| $mrr$ | 1.35, 1.40, 1.44, 1.50, 1.54, 1.60, 1.65, 1.69 |
| $mri$ | $10^{-8}$, 0.0005, 0.015, 0.07, 0.11, 0.3, 0.5 |

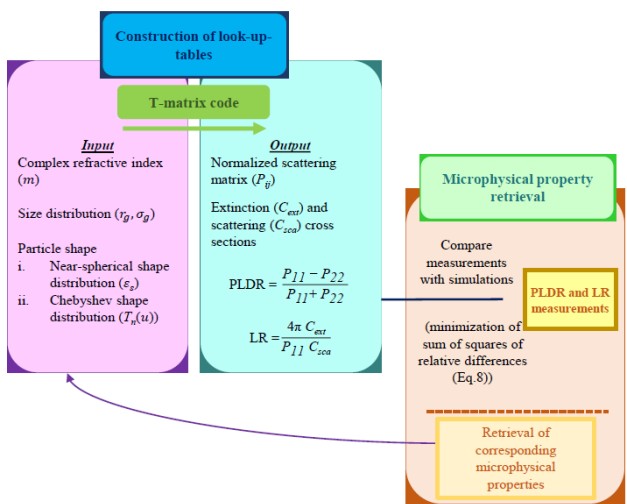

**Figure 1.** Overview of the methodology followed for the retrieval of the microphysical properties of the stratospheric smoke particles, using the PLDR and LR measurements at 355, 532 and 1064 nm: First, we construct appropriate look-up-tables of PLDR and LR values for near-spherical and Chebyshev particles using T-matrix calculations, and then we search in the look-up-tables for the solution that provides the best fit (minimization of Eq. (8)) of the PLDR and LR measurements.

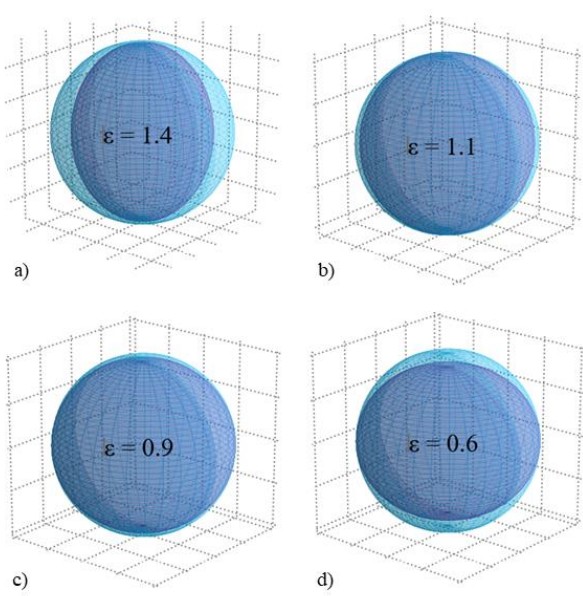

**Figure 2:** Examples of spheroids used (in dark blue colour), embedded in a perfectly spherical shell (in light blue colour), to visualize their deviation from the perfect sphere. Top row: prolate spheroids with (a) $\varepsilon = 1.4$ and (b) $\varepsilon = 1.1$ and bottom row: oblate spheroids with (c) $\varepsilon = 0.9$ and (d) $\varepsilon = 0.6$.

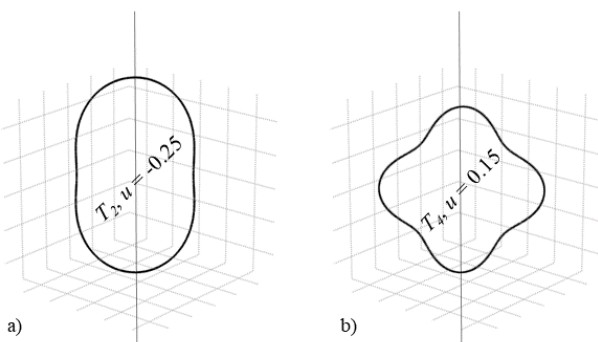

**Figure 3:** Examples of the Chebyshev particles used. From left to right: (a) Chebyshev particle of the second degree ($T_2$) with deformation parameter $u = -0.25$ and (b) Chebyshev particle of the fourth degree ($T_4$) with deformation parameter $u = 0.15$.

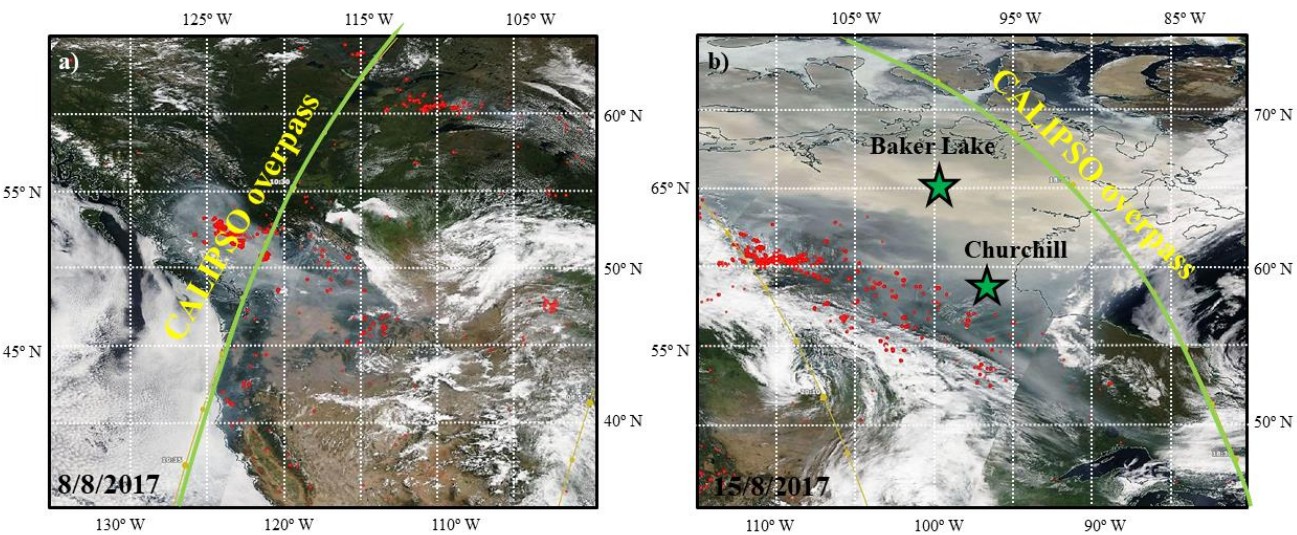

**Figure 4:** Corrected surface reflectance from MODIS, over-plotted with active fire regions and thermal anomalies (red dots) and CALIPSO ascending and descending overpasses (green lines). Red circles denote the position of the smoke plume on (a) 8 August2017 and (b) 15 August 2017. Green stars denote to stations located underneath the smoke plume that perform regular radiosonde measurements. Maps are generated from NASA Worldview Snapshots.

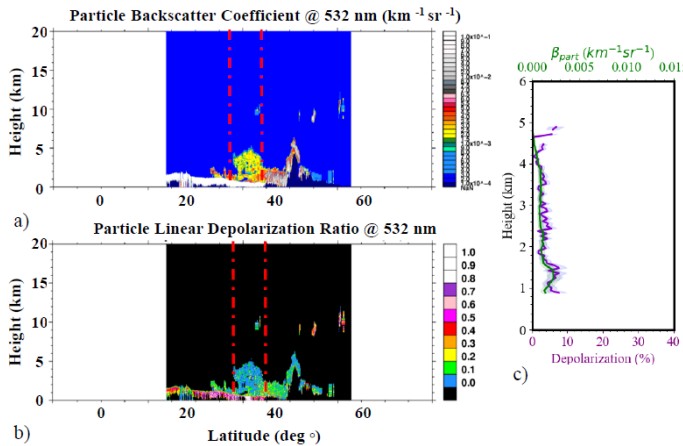

**Figure 5:** CALIPSO backscatter coefficient (km$^{-1}$sr$^{-1}$) and PLDR (%) that correspond to the night-time overpass on 8 August 2017, 10:27–10:41 UTC shown in Fig. 4a.  (a) The smoke plume is located between 39 and 45 ° latitude, below 6 km in altitude. Red dashed lines denote the spatial averaging applied for the retrieval of optical properties shown on the right plot.  (b) PLDR values at 532 nm, do not exceed values of 3–4 % at the height of the plume.

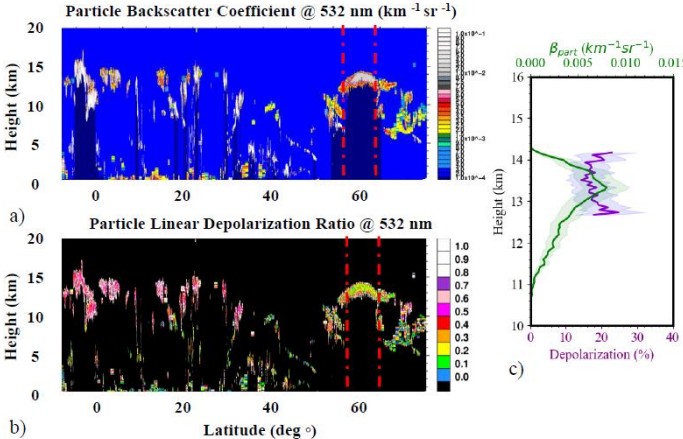

**Figure 6:** Same as Fig. 5 but for the day-time overpass of CALIPSO on 15 August 2017, 18:22–18:35 UTC shown in Fig.4b. (a) The smoke plume is now above the local tropopause at approximately 14 km, between 60 and 75 ° latitude. Red dashed lines denote the spatial averaging applied for the retrieval of optical properties shown on the right plot. (b) PLDR values at 532 nm (right plot, purple line) exceed 17% at the height of the plume. (Note that the altitude range for this plot is from 10 to 16 km, whereas in Fig. 5b is from 0 to 6 km).

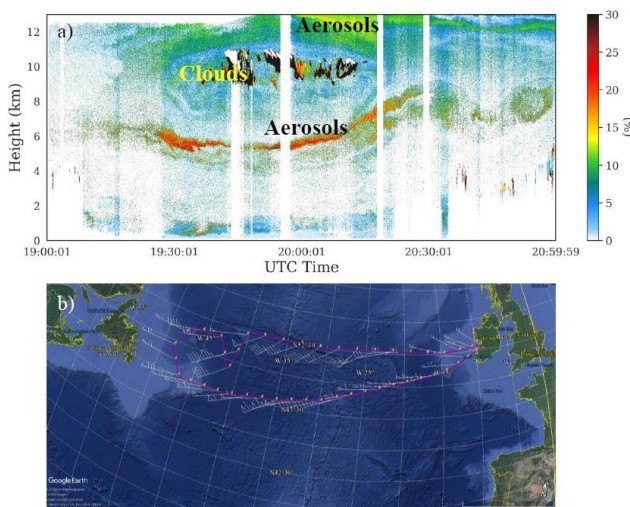

**Figure 7:** Time–height airborne lidar observations of the PLDR at 532 nm (a). Measurements were performed over the Atlantic Ocean, between 19:00 and 21:00 UTC on 7 October 2017 by the DLR "High Altitude and Long Range Research Aircraft" (HALO) in the framework of WISE mission. The track of the aircraft is shown in (b) over-plotted on Google Earth map.

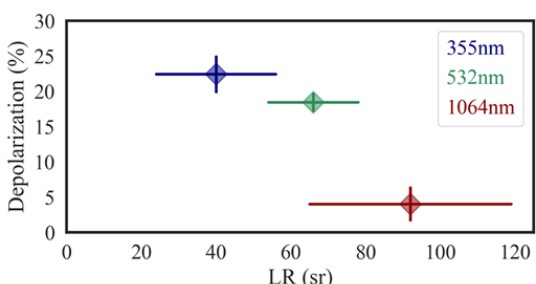

**Figure 8:** Intensive optical properties of the smoke particles found in the stratosphere, as measured on August 22, at Leipzig, Germany. The LR mean values are plotted against the PLDR mean values, along with the corresponding errors. A typically increasing behaviour of LR for aged Canadian smoke is observed at 355 and 532 nm, while for the PLDR the effect is the opposite: the surprisingly large, layer-integrated mean values drop from the UV to the Near-IR.

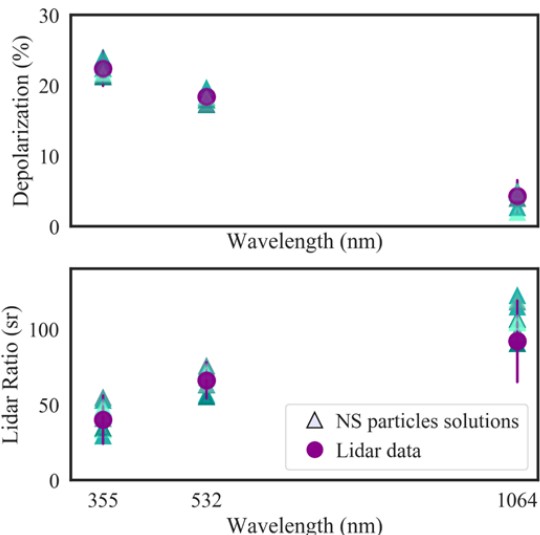

**Figure 9:** The reproduction of the measured PLDR and LR values, considering near-spherical particles. Purple circles correspond to measurements performed on 22 August 2017, at Leipzig, Germany, while purple lines correspond to the measurement uncertainties. Different light blue colour triangles denote to simulations performed with the T-matrix code, assuming near-spherical particles. Each light blue triangle corresponds to a different solution found to reproduce the measurements within their uncertainties, as given in Table 3. For these solutions the mean axial ratio $\varepsilon_s$ ranges from 1.1 to 1.4, the mean geometric radius $r_g$ ranges from 0.25 to 0.45 μm, and the wavelength-independent complex refractive index $m$, ranges from 1.35 to 1.55 for the real part $mrr$ and from 0.005 to 0.03 for imaginary part $mri$.

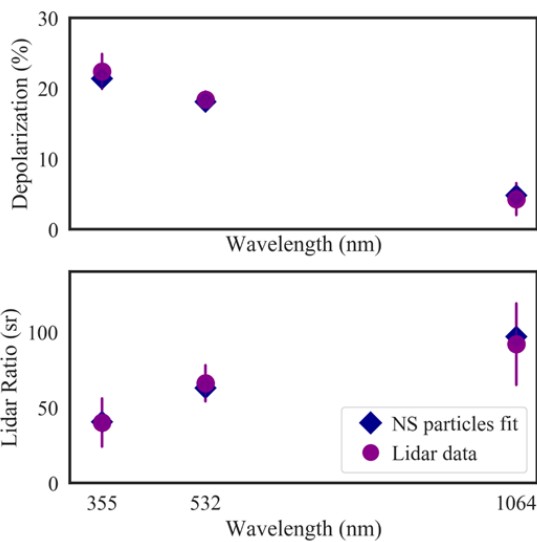

**Figure 10:** Same as Fig. 8 but only for the solution found to minimize the cost function of Eq. (8). Again, purple circles and lines correspond to measurements and measurements uncertainties on  22 August 2017, at Leipzig, Germany, while dark blue diamonds correspond to simulations assuming near-spherical particles of mean axial ratio $\varepsilon_s = 1.4$,  mean geometric radius $r_g = 0.25$ µm and a wavelength-independent complex refractive index $m = 1.55 + i0.025$ (this is the solution highlighted in blue in Table 3).

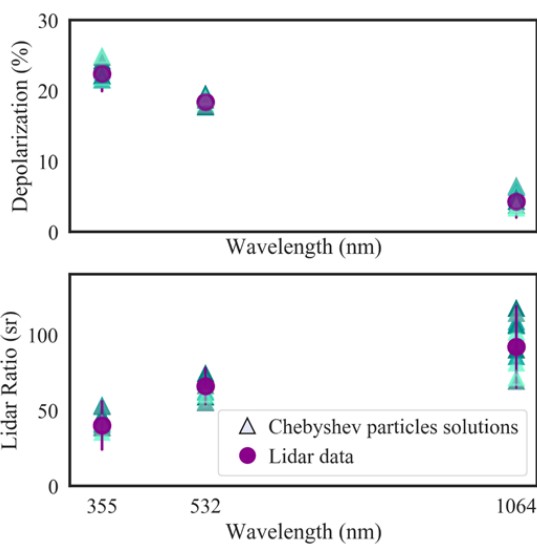

5    **Figure 11:** The reproduction of the measured PLDR and LR values, considering Chebyshev particles of the second degree ($T_2$). Purple circles correspond to measurements performed on 22 August 2017, at Leipzig, Germany, while purple lines correspond to the measurement uncertainties. Different light blue colour triangles denote to simulations performed with the T-matrix code, assuming Chebyshev particles of the second degree ($T_2$). Each light blue triangle corresponds to a different solution found to reproduce the measurements within their uncertainties, as given in Table 4. For these solutions, the deformation parameter $u$ ranges from - 0.25 to 0.15, the mean geometric radius

10    $r_g$ ranges from 0.2 to 0.5 μm, and the wavelength-independent complex refractive index $m$, takes values of 1.4 to 1.8 for real part $mrr$ and 0.015 to 0.055 for the imaginary part $mri$ .

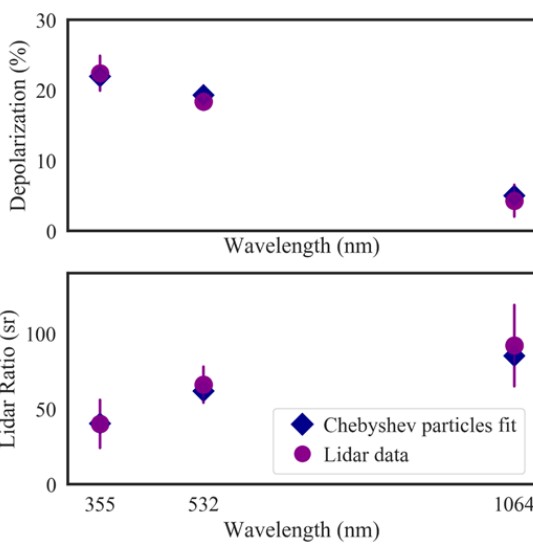

**Figure 12:** Same as Fig. 11 but only for the solution found to minimize the cost function of Eq. (8). Again, purple circles and lines correspond to measurements and measurement uncertainties on 22 August 2017, at Leipzig, Germany, while dark blue diamonds to simulations assuming Chebyshev particles of the second degree ($T_2$) that resemble oblate near-spherical particles, with deformation parameter $u = -0.25$, mean geometric radius $r_g = 0.2$ μm and a wavelength-independent complex refractive index $m = 1.65 + i0.03$ (this is the solution highlighted in blue in Table 4).

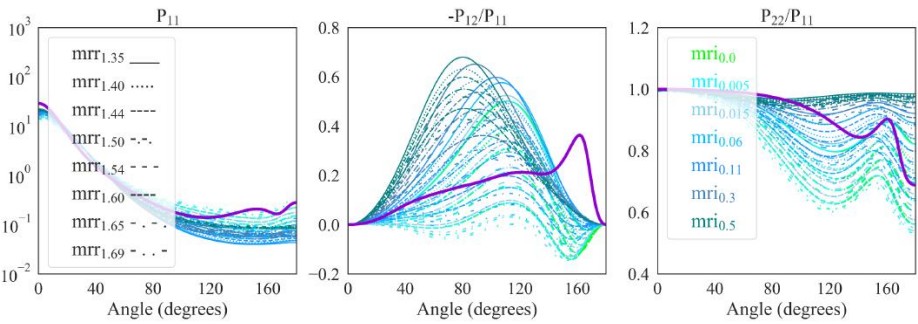

**Figure 13.** Comparison of the optical properties at $\lambda = 440$ nm for near-spherical particles (purple line) and the particles considered in the AERONET non-spherical model (blue lines). Left: $P_{11}$ (phase function), middle: $-P_{12}/P_{11}$ (degree of linear polarization), right: $P_{22}/P_{11}$. Purple lines in the plots: calculations considering the near-spherical particle properties derived for the stratospheric smoke particles from the Canadian fires, with mean axial ratio $\varepsilon_s = 1.4$, mono-modal, log-normal size distribution with $r_g = 0.25$ μm, $\sigma_g = 0.4$, and complex refractive index $m = 1.55 - i0.03$. Blue lines in the plots: calculations using the AERONET non-spherical model, mono-modal, log-normal size distributions with $r_g = 0.25$ μm and refractive indices of $mrr = 1.35, 1.40, 1.44, 1.50, 1.54, 1.60, 1.65, 1.69$ for the real part (different line styles in the plot) and $mri = 0.0, 0.005, 0.015, 0.06, 0.11, 0.3, 0.5$ for the imaginary part (different line colors in the plot).

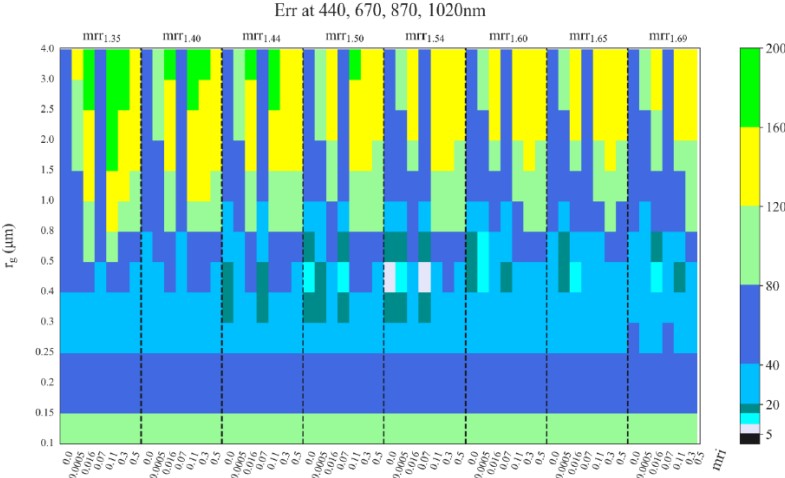

**Figure 14.** The residual error ($Err$) of fitting the phase functions at 440, 670, 870 and 1020 nm of the near-spherical particles presented in the manuscript, with the phase functions calculated with the AERONET non-spherical model for radius $r_g$ and complex refractive index $m$ shown in y- and x-axis, respectively.