# Peer review of "Is the near-spherical shape the "new black" for smoke?"

_Atmospheric Chemistry and Physics, 2020_

## Referee Comment (RC1) · Anonymous Referee #3 · 17 Mar 2020

General comments:

The more frequent occurrence of pyroCb smoke plumes lofted into the stratosphere is a rich target for analysis into a kind of aerosol about which we know relatively little. I'm particularly interested to see apparently successful modeling of three-wavelength values of both linear particle depolarization ratio and lidar ratio from lidar measurements of one such plume. This is a new result and potentially quite useful, since previous particle modeling studies did not have access to such a complete lidar observation and also because they, in general, resorted to much more complicated models than what the current authors have found to be useful. So, for this reason, primarily, I would like to see this paper published.

On the other hand, many aspects of the manuscript seem rather weak and unconvinc-

ing, so I believe major revisions are appropriate. First, the discussion and elimination of other models is not convincing. It may not be strictly necessary to show that other models perform worse if the near-spherical model is able to reproduce all the available measurements (because a simple model that fits all the observations has benefits over a more complicated model just by virtue of being simpler), but since an attempt is made to do so, it should be done thoroughly and correctly. Second, an unsubstantiated claim is made that this could improve AERONET retrievals. The idea of the new model improving or complementing AERONET is potentially quite appealing, and if done right this could be a major focus of the paper, so again it should be addressed thoroughly, not haphazardly. Finally, the speculation about the role of sulfuric acid for explaining the depolarization measurements seems a bit far-fetched and very difficult to validate while other potential explanations have not been adequately discussed. In my opinion, this is the weakest part of the manuscript and the best solution may be to simply not offer explanations for the shape at all, but rather to present this work as an advancement in the modeling of the optical properties alone. Otherwise, if the authors want to keep this, then a better, more thorough discussion of alternate theories and ways to distinguish between theories is needed.

Specific comments:

page 3, line 10, I think rather than "an explanation that could justify" the values, you're more fundamentally searching for "a model that can reproduce" the observations. This is a more precise statement of what this calculation is able to do and valuable enough at this stage.

page 3, line 10. "non-typical spectral dependence". What do you mean non-typical? Compared to what? It's my understanding that there are only a very small number of observations of three wavelengths of smoke particle linear depolarization ratio and not much discussion of two-wavelength observations. So, how do we know what spectral dependence is typical?

Besides just listing three papers at page 3, line 14, the introduction should discuss how this study is similar and different to the modeling studies in those and other papers, including those that use other modelled particle shapes other than near-spherical. Besides the 3 references listed, consider Kahnert (2017), Liu and Mishchenko (2018), Kanngießer and Kahnert (2018), Ceolato et al. (2018), and Luo et al. (2019) (some of these are mentioned later in the manuscript but not the introduction). Also Mishchenko et al. (2016) used multiple particle shapes, not just the near-spherical.

Kahnert, M.: Optical properties of black carbon aerosols encapsulated in a shell of sulfate: comparison of the closed cell model with a coated aggregate model, Optics Express, 25, 24579-24593, 10.1364/OE.25.024579, 2017.

Liu, L., and Mishchenko, M.: Scattering and Radiative Properties of Morphologically Complex Carbonaceous Aerosols: A Systematic Modeling Study, Remote Sensing, 10, 1634, 2018.

Kanngießer, F., and Kahnert, M.: Calculation of optical properties of light-absorbing carbon with weakly absorbing coating: A model with tunable transition from film-coating to spherical-shell coating, Journal of Quantitative Spectroscopy and Radiative Transfer, 216, 17-36, https://doi.org/10.1016/j.jqsrt.2018.05.014, 2018.

Ceolato, R., Gaudfrin, F., Pujol, O., Riviere, N., Berg, M. J., and Sorensen, C. M.: Lidar cross-sections of soot fractal aggregates: Assessment of equivalent-sphere models, Journal of Quantitative Spectroscopy and Radiative Transfer, 212, 39-44, https://doi.org/10.1016/j.jqsrt.2017.12.004, 2018.

Luo, J., Zhang, Q., Luo, J., Liu, J., Huo, Y., and Zhang, Y.: Optical Modeling of Black Carbon With Different Coating Materials: The Effect of Coating Configurations, Journal of Geophysical Research: Atmospheres, 124, 13230-13253, doi:10.1029/2019JD031701, 2019.

How are the ranges arrived at that are shown in Table 1, and also specifically the fixed

values for the distribution widths?

Can you eliminate water or ice cloud as an explanation for the measurements in CALIOP?

Figure 7 had a panel of water vapor mixing ratio but no explanation of this measurement or description of what impact this figure has on the analysis of this case.

For Figure 8 and Table 4 and others, please define the meaning of the error bars. Is this a true calculated uncertainty including both random and systematic error, or is this the standard deviation of available measurements, or something else? If it's standard deviation, how well do you think this captures the actual uncertainty of the lidar measurements at each wavelength? I ask because you mentioned that the 1064 nm extinction measurement is more challenging to make and we also know from literature that particle depolarization ratio in particular can be subject to significant systematic error in some circumstances (e.g. Burton et al. 2015, Freudenthaler 2016, Belegonte et al. 2018).

Freudenthaler, V.: About the effects of polarising optics on lidar signals and the △90Âăcalibration, Atmos. Meas. Tech., 9, 4181-4255, 10.5194/amt-9-4181-2016, 2016.

Belegante, L., Bravo-Aranda, J. A., Freudenthaler, V., Nicolae, D., Nemuc, A., Ene, D., Alados-Arboledas, L., Amodeo, A., Pappalardo, G., D'Amico, G., Amato, F., Engelmann, R., Baars, H., Wandinger, U., Papayannis, A., Kokkalis, P., and Pereira, S. N.: Experimental techniques for the calibration of lidar depolarization channels in EAR-LINET, Atmos. Meas. Tech., 11, 1119-1141, 10.5194/amt-11-1119-2018, 2018.

page 7, line 20-21. Lidar ratio increase from UV-visible suggests that it also increases from visible to near-IR. This should be deleted. There's nothing in Muller et al. (2007) that addresses the lidar ratio values in the near-IR one way or the other.

page 7, line 22 "far from typical". I urge you to reword and avoid "typical". Muller et

al. 2007 was a quite valuable paper, but the cases in it are somewhat limited, and it is now more than a decade old. Something that does not conform to Muller et al. 2007 is not necessarily "non-typical". We are still seeing new and different observations, by now including many of depolarizing smoke. I would say that this manuscript and other recent papers make a more convincing case that there is no "typical" for smoke or else that we do not have sufficient observations yet to know what is "typical", rather than that the depolarization ratios in this case are non-typical.

The notation in Equation 8 is confusing and doesn't really make sense. Please define what are i and n? Is this a summation over the three wavelengths? In that case, you have two subscripts (i and lambda) that mean the same thing? Or maybe i is binary and means lidar ratio and depolarization ratio, but in that case, you do not show how the different wavelength measurements are combined.

Equation 8 furthermore should arguably have the measurement uncertainty rather than the measurement itself in the denominator. This would be a more meaningful cost function considering you intend to compare the result to the measurement uncertainty in Eqn 9. Doing this could have a significant impact on your results, specifically the result for the Chebyshev model that is shown in Figure 12. The only simulated point that doesn't fit the measurements is the 532 nm depolarization which has a very small reported uncertainty and therefore not much tolerance. But if the cost function reflected the error bars as well, you might find there is a solution that fits that point at the expense of slightly larger discrepancy in another quantity where the uncertainty tolerance is much larger (e.g. lidar ratios).

I'm also curious how many solutions fit the criteria in Eqn 9 (or revised criteria) besides the minimum. Looking at this would give some insight into the uncertainty of your modeled results and the degree to which the set of measurements is sensitive to the complete set of free parameters in your model.

In the figure 10 comparison with AERONET, the use of a generic biomass burning solution instead of a solution for the same smoke plume seems like a needless shortcut that undercuts your ability to draw conclusions from it. I realize there were no precisely coincident AERONET measurements, but a previous paper studying the same event that you already reference by a coauthor (Haarig et al. 2019) shows an AERONET retrieval that is at least of the same smoke plume, and also apparently better agreement, so clearly it's possible to get better fidelity than the unrelated generic case given by Dubovik et al. (2002).

Furthermore, comparing a fit to a monomodal size distribution to the fit from a bimodal size distribution and then noting that the modes don't line up is not particularly useful, and it's not obviously tied to to the presence or absence of near-spherical particles per se. If you must compare a monomodal fit to a bimodal fit, then at least calculate the effective radius and variance (quantities that are more comparable from different distribution types) from each of them and compare that instead.

On a related note, could there be a coarse mode that your model is ignoring that might explain some of the features of your observations? Have you tried to eliminate the possibility of an optically significant coarse mode?

Fractal aggregates: Fractal aggregates require a lot of parameters to describe them, and some of them you held fixed instead of varying. If you cannot explore the full parameter space, how do you know that fractal aggregates can't fit the observations?

At line 25, you then point out that previous authors (not just Ishimoto et al. 2019, but see also Kahnert 2018, Kanngeisser and Kahnert 2018, Luo et al. 2019 and Ceolato et al. 2018, as noted above) have already established that bare aggregates are not able to reproduce measurements as well as coated aggregates, so I don't see a lot of value in running a model type that has already been shown not to work without also running a related model type that has been used with some success in the past (granted with less complete measurements in the past; indeed, the new measurements are the real strength of this contribution and where you have the opportunity to go beyond prior

work).

Page 7, line 10. This brief sentence about extending AERONET to include near-spherical particles is an interesting idea, but unsupported. To address this properly, please consider at least these 3 points. First, as stated above, there would have to be a fair comparison between AERONET results and your results for similar cases. As part of this, there would have to be an assessment of not just the size distribution, but also a reconstruction of the lidar measurements using the AERONET solution. Can you show that the AERONET retrieval fails to reproduce the lidar ratio and linear particle depolarization ratio adequately? Conversely, how does your near-spherical model do in reproducing AERONET radiances at all AERONET wavelengths? Whatever the answer to each of these questions, there's something to be learned. If the near-spherical model does a better job of reproducing lidar measurements and is also better at modeling AERONET measurements, then it could genuinely be an improvement for AERONET. If it doesn't improve the AERONET fits but improves the lidar fits, it might be less useful for AERONET alone (at least it would suggest that it might be hard for AERONET to operationally use the model if there is not sufficient measurement information content to distinguish near-spherical from spherical particle shapes), but might still be potentially of significant value for combined AERONET-lidar retrievals (e.g. constrained backscatter lidar retrievals). Even if the near-spherical model does a worse job at modeling AERONET measurements but a better job at modeling lidar measurements, it at least points us to the need for further modeling studies to find a single model that can unify both types of measurements.

page 7, line 16. See also Kablick et al. 2018 for another case discussing ice.

Kablick III, G., Fromm, M., Miller, S., Partain, P., Peterson, D., Lee, S., Zhang, Y., Lambert, A., and Li, Z.: The Great Slave Lake PyroCb of 5 August 2014: Observations, Simulations, Comparisons With Regular Convection, and Impact on UTLS Water Vapor, Journal of Geophysical Research: Atmospheres, 123, 12,332-312,352, 10.1029/2018jd028965, 2018.

page 7, line 17. In 2015, Burton et al. could not have extensively discussed studies that were published 3 or 4 years later. Specifically there's no discussion in Burton et al. 2015 about the ice hypothesis. It would be better for this manuscript's authors to take the opportunity to address these newer theories more completely here.

page 7, line 17-18. I'm not following the statement "soil lifting ...could explain the ....observations presented in this study". Do you mean to eliminate this possibility or support this possibility?

page 7, line 18. The Angstrom exponent is indeed confusing, primarily because we don't know what wavelengths you're referring to either in the measurements or in the comparison dataset that causes you to say this is "low". It's not uncommon for smoke measurements to have significant curvature in the spectral AOD (or extinction) (see Eck et al. 1999) and in fact Haarig et al. 2018 show a significant difference between the 355-532 nm and 532-1064 nm Angstrom exponents for their analysis of this smoke plume. Taking this into account, do you still believe the Angstrom exponent for this case indicates coarse mode particles? Please add a more complete discussion.

Eck, T. F., Holben, B. N., Reid, J. S., Dubovik, O., Smirnov, A., O'Neill, N. T., Slutsker, I., and Kinne, S.: Wavelength dependence of the optical depth of biomass burning, urban, and desert dust aerosols, Journal of Geophysical Research: Atmospheres, 104, 31333-31349, 10.1029/1999JD900923, 1999.

page 7, line 21 "surface roughness alterations"? I don't follow where this idea comes in the paper. Is this what the Chebyshev particle shape is meant to represent? There are other model representations of surface roughness (e.g. Liu et al. 2013, Kemppinen et al. 2015) so this label is probably too non-specific (vague) in this context and should be made more precise.

Liu, C., Lee Panetta, R., and Yang, P.: The effects of surface roughness on the scattering properties of hexagonal columns with sizes from the Rayleigh to the geometric optics regimes, Journal of Quantitative Spectroscopy and Radiative Transfer, 129, 169-

185, https://doi.org/10.1016/j.jqsrt.2013.06.011, 2013.

Kemppinen, O., Nousiainen, T., and Lindqvist, H.: The impact of surface roughness on scattering by realistically shaped wavelength-scale dust particles, Journal of Quantitative Spectroscopy and Radiative Transfer, 150, 55-67, https://doi.org/10.1016/j.jqsrt.2014.05.024, 2015.

page 10, line 2, what do you mean by "superposition" in this context? The papers above that address coatings (e.g. Kahnert 2017) suggest that the mixing of soot aggregates and coatings has a rather complicated impact on the optical properties, which I believe you would have to model rather explicitly and not just average, but perhaps the reference you're quoting suggests otherwise. Please describe more explicitly.

page 10, line 8 "two criteria need to be fulfilled". This seems like a spuriously specific conclusion. The lab study showed that the optical properties change in the presence of H2SO4, but does it follow that it must be H2SO4 and not other coating substances that are more likely to be found in a smoke plume? Or that there are not other mechanisms unrelated to this lab result that might explain the depolarization?

page 10, lines 15-20, the two hypotheses that the volcanic plume is required to explain the observations and that the smoke plume and volcanic plume intersected each seem like a stretch. There been several pyroCb plumes displaying elevated depolarization ratios in CALIPSO data, as well as the high tropospheric case described by Burton et al. 2015. Do you propose that they all intersected with volcanic plumes? If so, I guess it would be straightforward to check and you should do so. Also, the volcanic plume itself should be traceable with CALIPSO at least for part of its lifetime. Investigating this could help determine more clearly if the two plumes could have intersected at the same altitude and location.

Later on page 10, lines 29-30, is the suggestion that lower RH with aging of the plume decreased the depolarization ratio of the plume. This seems somewhat counter-intuitive, since with coated particles in the troposphere, at least, we would not expect

that losing a coating could make particles more spherical. Does the lab study support the idea that losing the coating makes the particles more spherical? Does the lab study, or your proposed follow-on, take the colder stratospheric temperatures into account?

Alternately, is there an indication that the size of the particles are changing (from losing the coating) for instance a change in the Angstrom exponents?

What results does your near-spherical model retrieval produce when applied to this later less-depolarized observation? Can you reproduce the reduction in linear particle depolarization ratio and also reproduce lidar ratio and extinction spectral dependence? If so, are the particle sizes smaller, or what other differences do you observe compared to the solution for the earlier time frame?

page 10, line 25, "ongoing research investigates whether this concurrence, or the the large amounts of SO2 "internally" released by the fire, or even the stratospheric sulfate background already pre-existing at the stratospheric smoke injection height, is the reason behind the unique large values of stratospheric PLDR." While you are obviously not required to describe your future research here in any detail, this teaser is so broad that it invites skepticism. I'd be interested to see it replaced with a more concrete description of what falsifiable question(s) you will attempt to answer and whether your proposed research is a lab study, based on in situ measurements, or a theoretical study.

Minor comments:

While informal titles can catch people's attention, I wonder if this one is really a good idea? "The new black" means "fashionable" or "popular", which is probably not quite what you're hoping for from your new smoke particle model. In general I appreciate funny titles but if it is a pun, I'm not getting it, since I don't see what scientific meaning "black" has in this title. Also, the phrase "the new black" is itself something of a fad which may fade quickly, leaving readers 5 or 10 years from now completely confused about what the title means. But of course it is up to the author; this isn't a comment

Interactive
comment

that needs a response, just a perspective on how one reader sees it, in case you find this helpful.

page 2, line 26. The sentence starting "implication for enrichment of smoke plumes with dust particles" should probably be deleted. Later on, you mention several possible explanations from previous literature, but here you only mention one without discussion that you later dismiss. Better to delete it here and hold the discussion until you are ready for it in the later section.

page 7, line 21 "not currently supported by other observation evidence found in the literature". Probably should be reworded. It seems like you're saying there is other evidence found in the literature that doesn't support your finding, but my understanding is that there is no contradictory observational evidence because lidar ratio at 1064 nm for this kind of observation has never been reported before! Please make this more clear. Having unique measurements is a real strength! No need to muddy the picture with imagined controversy.

page 7, line 11, probably delete the brief mention of pollen, which has not been addressed at all in this paper. We have no way of knowing whether the near-spherical model has any success in modeling pollen.

page 9, line 12, "results in near-spherical shapes" should be reworded. Logically, the finding that the near-spherical particle model reproduces a set of measurements better than a few other models could still be merely a very useful approximation. It does not necessarily mean the particles are literally shaped like Figure 2.

page 9, line 12, "previous studies" should be "some previous studies" (i.e. but not all, see for example Murayama et al. 2004, Burton et al. 2015 who specifically dispute it for certain other cases) Murayama, T., Müller, D., Wada, K., Shimizu, A., Sekiguchi, M., and Tsukamoto, T.: Characterization of Asian dust and Siberian smoke with multiwavelength Raman lidar over Tokyo, Japan in spring 2003, Geophys Res Lett, 31, 10.1029/2004gl021105, 2004.)

page 9, line 14, Sugiomoto et al. what year? (typo)

page 9, line 23, "advocate dissuasive towards" should be changed to (e.g.) "argues against"

page 9, line 26, "while up to now the LR values are not reproduced either". Again, I feel like this should be reworded, because it's making a confusing (or perhaps just incomplete) point when a much stronger one is indicated. Most previous modeling papers did not have the opportunity to reproduce three-wavelength lidar ratios because these observations have only just recently been published. The strength of the current manuscript is these new and unique measurements.

page 9 line 30. I think 15 micrometer monomers must be a typo.

page 10 line 4. "Thought" should be "Although"'

Table 4 caption. Please specify the instrument that made these measurements.

Figure 6. The red dashed lines appear to be in the wrong place.

It would be good to include a figure explaining what Chebyshev particles look like.

Figure 7. What do the white pixels near the top of the layer signify, in both the linear depolarization ratio and the water vapor mixing ratio?

Figure 7. What do the black down-arrows on the latitude and longitude axes represent?

Figure 7. Please indicate in 7c the portions of the track that are represented in 7a and 7b.

Figure 14. Please mark the location of the smoke plume and consider plotting this on an altitude (rather than pressure) scale and with altitude range more comparable to the ranges shown in Figures 6 and 7.

Please also consider putting an indicator of distance scale on Figures 6, 7, and 14.

---

## Short Comment (SC1) · 17 Mar 2020

In this note I offer comments limited to Section 3 and 5 regarding the evidence for mixing of volcanic sulfuric acid with smoke, for the authors to consider.

This manuscript makes an effort to suggest possible effects of mixing volcanic $H_2SO_4$ with smoke and thereby creating a "new black" particle linear depolarization ratio (PLDR). The example given is a volcanic $SO_2$ plume on 8 August 2017. In looking at the figures and text I see some apparent inconsistencies and possible misinterpretation. Figure 15 shows a map that includes upper tropospheric $SO_2$, and attributes that to Shiveluch. (The bottom panel of Figure 15 is confusing in that it refers to Himawari-9 imagery but the analysis illustrated is something else.) Although an eruption on 8 Au-

gust at Shiveluch was reported, it seems much more likely that the SO2 came from a 7 August 2017 eruption of Bogoslof in the Aleutians.

https://www.adn.com/alaska-news/2017/08/07/alaskas-tiny-bogoslof-volcano-erupts-again-sending-an-ash-cloud-miles-above-the-aleutians/

https://volcano.si.edu/volcano.cfm?vn=311300

It becomes apparent that the SO2 derives from this eruption when one follows the OMPS SO2 back in time. The plume starts on 7 Aug right near Bogoslof (credit NASA Worldview). Regardless of the source of the SO2, the only evidence suggestive of volcanic influence is the map of 8 August SO2 (Figure 15). The pyroconvection in Canada leading to the stratospheric smoke occurred 4 days later according to papers the authors cite. Presumably if volcanic sulfur was responsible for the months of double-digit PLDR "new black" one might expect to see a robust volcanic signature close to British Columba on or much closer to 12 August. What can be shown in that regard?

The paper refers to the 8 August CALIPSO measurement as "daytime" when it is in fact a night-time orbit segment. Consequently the connection made between this CALIPSO measurement and the daytime 8 August MODIS image in Figure 4 is inaccurate. On a technical note, the red dashed lines in Figure 6 for the 15 August CALIPSO data are not where the text directs the reader: the 13 km smoke layer.

The authors refer to a 12 August CALIPSO measurement (Page 6, line 23) but don't show any such measurement. It is apparent they meant 15 August but this needs to be clarified (if indeed they intend to show a 12 August measurement) or corrected.

On page 6, line 16 the authors seem to state that between 8 and 15 August the stratospheric smoke plume had already blown to Europe: "...8 and 15 August 2017, when the smoke plume has already reached Europe" They do not present any data to support that and I believe there is no support for that claim. The leading edge of the plume on 15 August was still entirely over Canada.

[Figure]

In summary, I was confused by the material in Section 3 and 5 and thus was left unconvinced of any meaningful mingling of volcanic sulfates and pyroCb-injected stratospheric smoke. Presumably, if the transport pathway was what the authors claim—pyroconvection—the sulfates would have to have been in large concentration in the vicinity of the pyroCbs and in the inflow part of the atmosphere (i.e. lower troposphere). If the two mingled by virtue of UTLS sulfates in high concentration encountering the pyroCb outflow, one might expect that the sulfates would be detectable leading up to the pyroCb injection. This might be an avenue for the authors to explore because there is good CALPSO coverage of the Canadian and upstream environments on all the days between 8 and 12 August.

It is becoming increasingly evident that double-digit PLDR is quite common for stratospheric smoke. In personal communication with one of the coauthors, I discussed a similar phenomenon in northern summer 2014. Here is an example of double-digit PLDR of stratospheric smoke over Scandinavia at that time (credit: http://lidar.ssec.wisc.edu/)

http://hsrl.ssec.wisc.edu/by_site/18/2014/08/17/am/#MF2HSRL

The Black Saturday (Australia, February 2009) pyroCb stratospheric smoke also had double-digit PLDR. Here is an example of week-old smoke at that time (Credit: NASA):

https://www-calipso.larc.nasa.gov/products/lidar/browse_images/show_detail.php?s=production&v=V4-10&browse_date=2009-02-15&orbit_time=12-52-14&page=3&granule_name=CAL_LID_L1-Standard-V4-10.2009-02-15T12-52-14ZN.hdf

It is unlikely, or at least un-established, that precursor volcanic activity occurred in these 2009 and 2014 cases. Hence it would seem that there is another common bond, albeit still unresolved, embodied in this growing record of anomalously large PLDR in dry stratospheric smoke environments.

---

## Referee Comment (RC2) · Anonymous Referee #2 · 19 Mar 2020

The authors use an example of transported stratospheric smoke from the 2017 Canadian wildfire pyroCB to model the spectral dependence of depolarization and lidar ratio at 355, 532, and 1064 nm. Near-spherical, Chebyshev, and fractal shapes are used, but only the near-spherical shapes are able to match the results obtained from lidar measurements.

The subject is quite relevant and the results could be very significant for scientific community. However, there are some issues with the manuscript which must be clarified before it is suitable for publication. Examples are provided below:

Firstly, the title implies that the near spherical shape may be the "new black" for smoke. However, the case study focuses on a stratospheric smoke case. It follows to ask if this is only the "new black" for smoke in the stratosphere only or should we assume this

[Figure]

might apply to the troposphere also? From the example shown, my guess is no.

Page 3, Line 10: Is the spectral dependence really non-typical? The authors do a good job of convincing us that the high depolarization values are non-typical for smoke, but spectral dependence seems the opposite. In fact, decreasing depolarization with increasing wavelength is closer to typical from the references cited by the authors.

Page 3, Line 12: Something is missing from this sentence or perhaps the wording is intended to be: The starting point and main assumption of our investigation is that the particle near-spherical-shape can be highly depolarizing as shown in the work of Mishchenko and Hovenier (1995); Mishchenko et al. (2016); Bi et al. (2018) and Ishimoto et al. (2019).

Figure 2: The images seem to be incorrectly placed, given the aspect ratios. Oblate sphere should be flattened and prolate stretched

Page 6, Line 20 - 26 and Figure 6: The latitude limits in the text do not match those in the Figure and the red dashed lines do not match the section highlighted in Figure 4. But even more importantly, the corresponding browse images indicate a complex mixture of aerosol and ice from cirrus clouds which would explain the high depolarization ratios in these cases.

https://www-calipso.larc.nasa.gov/products/lidar/browse_images/show_detail.php?s=production&v=V4-10&browse_date=2017-08-15&orbit_time=17-55-33&page=3&granule_name=CAL_LID_L1-Standard-V4-10.2017-08-15T17-55-33ZD.hdf

https://www-calipso.larc.nasa.gov/products/lidar/browse_images/show_detail.php?s=production&v=V4-10&browse_date=2017-08-15&orbit_time=17-55-33&page=4&granule_name=CAL_LID_L1-Standard-V4-10.2017-08-15T17-55-33ZD.hdf

The images from Haarig et al 2018 are more convincing for the case the authors are making. Perhaps another CALIPSO image should be used if the authors would like to show satellite measurements.

Page 8, Line 12-13: The authors state: What is interesting here is that the retrieved sizes for near-spherical smoke particles are absent in the AERONET climatology product. However, the difference in using a mono-modal versus bi-modal distribution is something that should be explored or at least discussed more in the text. Are we seeing the result of an "effective" radius in a mon-modal distribution that presents a possible solution for the high depolarization and spectral dependence, when there could also be another solution obtained from a mixture in a bi-modal distribution?

Sections 4.2 and 4.3: Is this all that can be said for these solutions... we tried these shapes and they didn't work? It would seem that more could be gleaned from these failed attempts. For instance, these shapes show the correct trend in spectral dependence even though the absolute values/relative differences do not fit the measurements.

Page 9, Line 18: Can the authors offer an explanation for the low angstrom exponents other than coarse particles?

---

## Referee Comment (RC3) · Anonymous Referee #1 · 22 Apr 2020

Review of "Is the near-spherical shape the new black for smoke?" by A. Gialitaki et al., 2020

The manuscript topic fits well within the journal scope is providing new insights on biomass burning aerosol layers. Nevertheless, it needs major revisions before being ready for publication.

Major comments:

1) A substantiated and consolidated verification of the measurement quality and the potential role of systematic errors affecting the measurements is a preliminary paramount step when such high PLDR values are measured. This is particularly true for stratospheric aerosols as calibration of aerosol depolarization measurements of strato-

spheric particles is quite difficult and cannot rely on molecular calibration approach.

2)The fact that such high PLDR values were reproduced using T-matrix simulations, assuming near-spherical shapes, for biomass burning is not itself a verification of the fact that observed particles were indeed transported stratospheric smoke plumes. More information on possible particle composition, and its possible organic origin, should be inferred from other optical measurements (multi-wavelength particle extinction and backscatter measurements).

3) Because of 2) the proposed approach is rater weak. It is not possible to generalize statements just from a single case study. Moreover it seems a sort of ill-posed problem and the minimum in Eq. 8 might be relative, i.e. what happens if instead a mono-modal distribution a bimodal is chosen? or a gamma instead of normal distribution? Probably Eq. 8 will provide independently a solution.

4) As stated by Sassen and Khvorostyanov, smoke can directly act as ice nuclei before liquid clouds form (https://iopscience.iop.org/article/10.1088/1748-9326/3/2/025006). This fact can partially explain the higher PLDR (considering a process in progress). This aspect, very likely is not mentioned in the manuscript and can be the reason of PLDR increase.

5) The simulations themselves are not original as in fact similar simulations were performed in the past by Bi et al. 2018, Mishchenko et al. 2016; Ishimoto et al., 2019, as the authors explicitly admit. What is different with respect to those manuscript?

6) The title (i found it funny) might be misinterpreted and considered inappropriate

Specific comments are available in the attached manuscript

Please also note the supplement to this comment:
https://www.atmos-chem-phys-discuss.net/acp-2020-22/acp-2020-22-RC3-supplement.pdf

[Figure]

[Figure]

**Supplement:**

[revised manuscript text omitted]

---

## Author Comment (AC1) · 16 Jul 2020

**General comments:**

The more frequent occurrence of pyro-Cb smoke plumes lofted into the stratosphere is a rich target for analysis into a kind of aerosol about which we know relatively little. I'm particularly interested to see apparently successful modeling of three-wavelength values of both linear particle depolarization ratio and lidar ratio from lidar measurements of one such plume. This is a new result and potentially quite useful, since previous particle modeling studies did not have access to such a complete lidar observation and also because they, in general, resorted to much more complicated models than what the current authors have found to be useful. So, for this reason, primarily, I would like to see this paper published. On the other hand, many aspects of the manuscript seem rather weak and unconvincing, so I believe major revisions are appropriate.

**REPLY:** We thank very much Reviewer #3 for his/her careful reading, comments and suggestions, which we address in the following. With his/her suggestions, we believe that the new version of the manuscript is significantly improved, and our findings are promoted in a better way. The author's answers along with the changes in the manuscript are listed below.

Remark: The figure numbers and the page numbers in the referee comments and in our replies correspond to the original manuscript.

First, the discussion and elimination of other models is not convincing. It may not be strictly necessary to show that other models perform worse if the near-spherical model is able to reproduce all the available measurements (because a simple model that fits all the observations has benefits over a more complicated model just by virtue of being simpler), but since an attempt is made to do so, it should be done thoroughly and correctly.

**REPLY:** We thank the reviewer for this comment. We added a more extensive discussion in the introduction on how our study differentiates for previous studies. Please see specific comments 3, 10 and 14 for the detailed changes according to this comment.

Second, an unsubstantiated claim is made that this could improve AERONET retrievals. The idea of the new model improving or complementing AERONET is potentially quite appealing, and if done right this could be a major focus of the paper, so again it should be addressed thoroughly, not haphazardly.

**REPLY:** We thank the reviewer for this very constructive comment. We improved the manuscript following the Reviewer's suggestions. Please refer to response for specific comment 13.

Finally, the speculation about the role of sulfuric acid for explaining the depolarization measurements seems a bit far-fetched and very difficult to validate while other potential explanations have not been adequately discussed. In my opinion, this is the weakest part of the manuscript and the best solution may be to simply not offer explanations for the shape at all, but rather to present this work as an advancement in the modeling of the optical properties alone. Otherwise, if the authors want to keep this, then a better, more thorough discussion of alternate theories and ways to distinguish between theories is needed.

**REPLY:** We thank the reviewer for this comment. At the current stage the discussion on the possible physical explanation for the smoke PLDR values is removed from the manuscript. More efforts will be made to investigate this issue and re-assess the possible coexistence of smoke particles and particles of sulfuring nature and whether this could affect the former in such a way to form near-spherical particles.

**Specific comments:**

**1.** page 3, line 10: I think rather than "an explanation that could justify" the values, you're more fundamentally searching for "a model that can reproduce" the observations. This is a more precise statement of what this calculation is able to do and valuable enough at this stage.

**REPLY:** Thank you for this comment, please find our answer below.

**2. page 3, line 10.** "non-typical spectral dependence". What do you mean non-typical? Compared to what? It's my understanding that there are only a very small number of observations of three wavelengths of smoke particle linear depolarization ratio and not much discussion of two-wavelength observations. So, how do we know what spectral dependence is typical?

**REPLY:** We thank the reviewer, we rephrased the following paragraph in order better position our research and to highlight the two comments above (**page 3**, **line 9**):

In contrast to prior studies, for our investigation for the stratospheric smoke originating from the Canadian wildfires, we do not adopt morphologically complex shapes of bare or coated smoke aggregates, which are

associated with excessive computations. Instead, we propose a much simpler model of compact near-spherical particles. Our starting point and main assumption is that the particle near-spherical-shape can be highly depolarizing, as shown in the work of Mishchenko and Hovenier (1995) and Bi et al. (2018). Our analysis shows that for the Canadian stratospheric smoke observed above Europe in August 2017, the PLDR and LR measurements along with their spectral dependence, can be successfully reproduced with the proposed model of compact near-spherical particles. The size and refractive index of the particles are estimated as well, and seem to agree well with past observations for aged smoke. We further examine the capability of this model to be used on an operational level and in particular as an extension to the AERONET operational aerosol retrieval (Dubovik et al., 2006), since it provides a much simpler and faster solution with respect to more complicated shapes for stratospheric smoke particles (e.g. Mishchenko et al., 2016; Ishimoto et al., 2019)."

3. Besides just listing three papers at page 3, line 14, the introduction should discuss how this study is similar and different to the modeling studies in those and other papers, including those that use other modelled particle shapes other than near-spherical. Besides the 3 references listed, consider Kahnert (2017), Liu and Mishchenko (2018), Kanngießer and Kahnert (2018), Ceolato et al. (2018), and Luo et al. (2019) (some of these are mentioned later in the manuscript but not the introduction). Also Mishchenko et al. (2016) used multiple particle shapes, not just the near-spherical.

**REPLY:** Thank you for your comments, an updated discussion has been included in the introduction to point how our study is different from the previous work. We specifically added the following paragraph (**page 2**, **line 25**):

"In the past, many studies have used simpler or more complicated particle shape models in order to reproduce the lidar measurements of smoke and provide a physical insight on light interaction with these particles. In Kahrent (2017), the PLDR of black carbon aggregates covered by a cell of sulphates was simulated by two different models; a closed cell (i.e. each monomer in the aggregate is coated separately) and a coated aggregate model (i.e. the whole aggregate is coated). Their analysis showed that for thicker coating the coated cell model of volume equivalent radius of 0.3 to  $0.4\mu$ m, can provide PLDR values of the order of 15% at 532nm. Mishchenko et al. (2016) and Liu and Mishchenko (2018) used rather complex morphologies for smoke particles, in order to reproduce the PLDR values measured by Burton et al. (2015). Amongst others, these morphologies included a) a fractal aggregate partially embedded in a spherical sulphate cell, b) two-externallymixed spherical sulphate cells, each hosting an aggregate (models 6 and 11 in Fig. 1 in Liu and Mishchenko, 2018) and c) a high-density aspherical soot core, encapsulated in a circumscription spheroid cell (with axial ratio of 0.9 to 1.2; model 4 in Fig. 2 in Mishchenko et al., 2016). All these morphologies reproduced successfully the smoke optical properties measured by Burton et al. (2015). Moreover, Luo et al. (2019) used twenty different configurations of coated fractal aggregates and showed that for relatively small fractal dimension (i.e. relatively fresh aggregates), and for small black carbon fractions (i.e. densely coated aggregates; configuration C in Fig. 2 in Luo et al. (2019)), the PLDR values can reach up to 40, 15 and 6% at 355, 532 and 1064nm, respectively. Ishimoto et al. (2019) used fractal aggregates and artificial surface tension induced on the particles to mimic the effect of coating by water soluble materials forming around the particles. This particular study present results for both the PLDR and the lidar ratio (LR), which is indicative of the composition of the particles. In Liu and Mishchenko (2019), tar ball aggregates were used to model exceptionally strong PLDR as those measured by Burton et al. (2015). The aforementioned studies highlighted the fact that that in order to reproduce significant PLDR values (higher than 20% at 532nm), the fractals need to be coated (i.e. shapes of "Type-B, size 11, Vr = 20" shown in Fig. 4 of Ishimoto et al. 2019). We should point out though that most of them refer to monodispersed particles, and averaging over size could possibly supress some of the observed features.".

**References** (that are not included in the initial version of the manuscript):

Kahnert, M.: Optical properties of black carbon aerosols encapsulated in a shell of sulfate: comparison of the closed cell model with a coated aggregate model, Optics Express, 25, 24579-24593, 10.1364/OE.25.024579, 2017.

Kanngießer, F., and Kahnert, M.: Calculation of optical properties of light-absorbing carbon with weakly absorbing coating: A model with tunable transition from film-coating to spherical-shell coating, Journal of Quantitative Spectroscopy and Radiative Transfer, 216, 17-36, https://doi.org/10.1016/j.jqsrt.2018.05.014, 2018.

Liu, L., and Mishchenko, M.: Scattering and Radiative Properties of Morphologically Complex Carbonaceous Aerosols: A Systematic Modeling Study, Remote Sensing, 10, 1634, 2018. Luo, J., Zhang, Q., Luo, J., Liu, J., Huo, Y., and Zhang, Y.: Optical Modeling of Black Carbon with Different Coating Materials: The Effect of Coating Configurations, Journal of Geophysical Research: Atmospheres, 124, 13230-13253, doi:10.1029/2019JD031701, 2019.

4. How are the ranges arrived at that are shown in Table 1, and also specifically the fixed values for the distribution widths?

**REPLY:** The fixed width of the size distribution is a simplification we used in order to reduce the retrieval complexity, considering that this parameter does not greatly affect the lidar-derived optical properties (e.g. Burton et al., 2016). We specifically chose the fixed  $\sigma_g = 0.4$  to represent a moderately wide size distribution. The fixed width of the shape distribution  $\sigma_s$  is also necessary for the reduction of the retrieval complexity. A small value of this width is used to avoid the wash-out of the characteristic optical properties of near-spherical particles which are shown for a relatively narrow aspect ratio range (e.g. Bi et al., 2018).

For the other microphysical properties used as inputs for the T-matrix calculations, the ranges were selected based on the ranges found in the literature for smoke particles. More specifically, in Muller et al. (2005) for aged Canadian and Siberian smoke, values of effective radius *reff* from 0.16 (± 0.04) to 0.41 (± 0.14)µm, and real part of the refractive index  $m_r$  values of 1.37 (± 0.04) to 1.65 (± 0.03) were retrieved from 3b + 2a lidar data inversion. In a following study by Muller et al. (2007) the *reff* of one-day-old smoke plumes was found to be 0.13 (± 0.04)µm, while for 18-day-old smoke plumes, the *reff* was much larger, equal to 0.37 (± 0.06)µm. Nicolae et al. (2013), combined lidar measurements and mass spectroscopy for 3/4-day-old SW Romanian smoke, and derived an *reff* of 0.19 (± 0.11)µm, while for 3-old Ukraine smoke the corresponding value was 0.40 (± 0.12)µm. For the same case studies, the complex refractive index values spanned from m = 1.41 (± 0.07) + i0.005 (± 0.003) to m = 1.66 (± 0.09) + i0.05 (± 0.01). In Giannakaki et al. (2015) for South Africa smoke, the *reff* values derived from lidar data inversion at 355nm spanned from 0.11 to 0.28µm, and the complex refractive index values were derived to be m = 1.43 (± 0.07) + i0.016 (± 0.01). In Dubovik et al. (2002), the climatological mean value of the complex refractive index derived from AERONET measurements for tropospheric smoke in the United States and Canada is m = 1.5 (± 0.05) + i0.0094 (± 0.003).

To cover the range of the reported values in the studies listed above, in the initial version of the manuscript we used ranges shown in Table.1. For the revised version, we further extended the refractive index values as follows:  $m_r = 1.35 - 1.85$  and  $m_i = 0.005 - 0.55$ .

We included the information above in the revised manuscript (page 3, lines 24 - 26 and page 18, Table 1). **References** (*that are not included in the initial version of the manuscript*):

Burton, S. P., Chemyakin, E., Liu, X., Knobelspiesse, K., Stamnes, S., Sawamura, P., Moore, R. H., Hostetler, C. A., and Ferrare, R. A.: Information content and sensitivity of the  $3\beta + 2\alpha$  lidar measurement system for aerosol microphysical retrievals, Atmos. Meas. Tech., 9, 5555–5574, https://doi.org/10.5194/amt-9-5555-2016, 2016

**5. Can you eliminate water or ice cloud as an explanation for the measurements in CALIOP?**

**REPLY:** CALIOP measurements at the northeastern Canada on 15 August 2017 (https://www-calipso.larc.nasa.gov/products/lidar/browse\_images/show\_v4\_detail.php?s=production&v=V4-

10&browse\_date=2017-08-15&orbit\_time=17-55-33&page=3&granule\_name=CAL\_LID\_L1-Standard-V4-10.2017-08-15T17-55-33ZD.hdf), show the stratospheric smoke layer at 11-14.5 km, where radiosonde measurements show temperatures below -40°C (Fig. 1 below). The radiosonde temperature profiles are from three stations close to the position of the smoke plume (Fig. 4b in manuscript): Churchill (Lat: 58.73, Lon: -94.08), Inukjuak (Lat: 58.45, Lon: -78.11) and Baker Lake (Lat: 64.31, Lon: -96.00). These low temperatures should exclude the presence of water clouds from CALIOP data, since even without the presence of aerosol particles, at these temperatures water can freeze homogeneously (Wallace and Hobbs, 2006; Fig. 6.29). Moreover, the ground-based lidar measurements on 23 August 2017 at Leipzig, show the stratospheric smoke layer at 14-16 km, where radiosonde measurements from the closest station (Lindenberg) provide temperatures below -50°C. Again the low temperatures indicate the absence of water clouds.

Regarding ice formation, the CALIOP PLDR values are below 20% both for the aforementioned overpass and for the closest overpass from Leipzig on 23 August 2017 (https://www-calipso.larc.nasa.gov/products/lidar/browse\_images/show\_v4\_detail.php?s=production&v=V4-10&browse\_date=2017-08-23&orbit\_time=01-29-01&page=1&granule\_name=CAL\_LID\_L1-Standard-V4-10.2017-08-23T01-29-01ZN.hdf (~90 km away and approximately 1 hour after the end of the ground based

lidar measurements reported from Haarig et al., 2018), while the attenuated color ratio (i.e., the ratio of particle backscatter coefficient at 1064nm) is below 1. Further analysis of CALIOP data provides a mean (median) value of the backscatter related Angstrom exponent (BAE) at 532/1064nm, of 0.9 (0.9) with a standard deviation on 1.07. For PLDR, typical values for cirrus clouds are usually no less than 40% (Chen at al., 2002; Noel et al., 2002; Voudouri et al., 2020) and the color ratio is expected to be close to 1 due to the large size of ice crystals compared to the lidar wavelengths. For the BAE, values close to zero are expected, although the CALIPSO data are highly noisy at these altitudes as indicated by the standard deviation.

Moreover, for the overflight close to Leipzig on 23 August 2017, the lidar ratio (LR) measured from the ground based system is  $(66 \pm 12)$  sr at 532nm. This is similar to the LR observed in the past for aged smoke particles (i.e. Fiebig et al., 2002; Veselovskii et al., 2015; Burton et al., 2012) but quite high for cirrus clouds (Gouveia et al., 2017).

Based on the above, although we cannot exclude the possibility of small ice crystals formed inside the stratospheric plume, we believe that the aforementioned characteristics indicate that this is not an ice cloud but rather a large smoke plume. Similar questions were raised also from Anonymous Reviewer 1, and Anonymous Reviewer 2. Hence we added the following paragraph to the manuscript to address the reviewers comment (**page 6, line 24**):

"Owning to the altitude of the smoke plume, one could attribute such PLDR values to the beginning of ice formation. Indeed, radiosonde temperature profiles from three stations located underneath the smoke plume (green stars on Fig.3b), reveal that the temperature above 11 km drops below -40C, at which point homogeneous ice formation can occur (Wallace and Hobbs, 2006). However, the PLDR values of cirrus clouds are usually no less than 40% (Chen at al., 2002; Noel et al., 2002; Voudouri et al., 2020). Further analysis of CALIOP data provides a mean (median) value of the backscatter related Angstrom exponent (BAE) at 532/1064nm of 0.9 (0.9) with a standard deviation on 1.07. For the BAE values close to zero are expected for cirrus clouds, although, as indicated by the large standard deviation, CALIPSO data are highly noisy at these altitudes. A recent study by Yu et al. (2019) also showed that the largest fraction of stratospheric smoke particles consisted of organic carbon (98% compared to 2% for black carbon). Particles of such high organic carbon content serve poorly as ice nuclei (Kanji et al., 2017; Phillips et al., 2013). Although the possibility of

small ice crystals formed inside the smoke layers cannot be excluded, (largely due to the absence of in situ measurements) the aforementioned characteristics indicate that this plume consists of smoke particles rather than ice crystals."

**Figure 1.** Corrected surface reflectance from MODIS on 15 August 2017, over-plotted with the PyroCb aerosol index product from Suomi NPP/OMPS (in yellow). Green stars indicate the position of the radiosonde stations used, while green line marks the CALIPSO overflight during 18:22 – 18:35 UTC. Maps are generated from the NASA Worldview Snapshots.

---

## Author Comment (AC2) · 16 Jul 2020

*The authors use an example of transported stratospheric smoke from the 2017 Canadian wildfire pyroCB to model the spectral dependence of depolarization and lidar ratio at 355, 532, and 1064 nm. Near-spherical, Chebyshev, and fractal shapes are used, but only the near-spherical shapes are able to match the results obtained from lidar measurements. The subject is quite relevant and the results could be very significant for scientific community. However, there are some issues with the manuscript which must be clarified before it is suitable for publication. Examples are provided below:*

**REPLY**: Thank you very much for your helpful comments. Please find our point-by-point response below.

Remark: The figure numbers and the page numbers in the referee comments and in our replies correspond to the original manuscript

*1. Firstly, the title implies that the near spherical shape may be the "new black" for smoke. However, the case study focuses on a stratospheric smoke case. It follows to ask if this is only the "new black" for smoke in the stratosphere only or should we assume this might apply to the troposphere also? From the example shown, my guess is no.*

**REPLY:** Thank you for this question. To the best of our knowledge, up to now the majority of the cases reported for smoke particle linear depolarization ratio (PLDR) approximating 20% at 532nm, refer to smoke found in the stratosphere. The sole exception is the case study reported in Burton et al. (2015) for a smoke plume found at 8km height. All the cases were associated with PyroCb activity and all are indicative of high depolarization values in both troposphere and stratosphere, this is why we didn't separate in the title. We make sure we mention this throughout the revised manuscript and we further included the following to make this more obvious to the reader (**page 7, line 24**):

"To the best of our knowledge, up to now the majority of observations for such smoke PLDR values, refer to smoke particles found in the stratosphere (i.e. Ohneiser et al., 2020). The sole exception is the case study reported by Burton et al. (2015) (see also Table 2). "

*2. Page 3, Line 10: Is the spectral dependence really non-typical? The authors do a good job of convincing us that the high depolarization values are non-typical for smoke, but spectral dependence seems the opposite.*

*In fact, decreasing depolarization with increasing wavelength is closer to typical from the references cited by the authors.*

**REPLY:** We agree with the reviewer that this statement should be reworded, primarily because there is still limited amount of information on such case studies for smoke. The only cases that up to now have reported observations at three lidar wavelengths are Haarig et al. (2018), Hu et al. (2019) and Burton et al. (2015). The first two refer to the same case of British Columbia fires of 2017, while the last one refers to the Pacific Northwest fires of 2014. There is a notable difference in PLDR values at 532nm reported from Haarig and Hu (~18%), compared to those reported from Burton (~9%), but still this may not be sufficient information to characterize the spectral dependence as non-typical.

The following has been re-worded in the manuscript in order to highlight this comment (**page 3, line 10**): "In contrast to prior studies, for our investigation for the stratospheric smoke originating from the Canadian wildfires, we do not adopt morphologically complex shapes of bare or coated smoke aggregates, which are associated with excessive computations. Instead, we propose a much simpler model of compact near-spherical particles. Our starting point and main assumption is that the particle near-spherical-shape can be highly depolarizing, as shown in the work of Mishchenko and Hovenier (1995) and Bi et al. (2018). Our analysis shows that for the Canadian stratospheric smoke observed above Europe in August 2017, the PLDR and LR measurements along with their spectral dependence, can be successfully reproduced with the proposed model of compact near-spherical particles."

*3. Page 3, Line 12: Something is missing from this sentence or perhaps the wording is intended to be: The starting point and main assumption of our investigation is that the particle near-spherical-shape can be highly depolarizing as shown in the work of Mishchenko and Hovenier (1995); Mishchenko et al. (2016); Bi et al. (2018) and Ishimoto et al. (2019).*

**REPLY:** Thank you, we have rephrased the following in the revised manuscript (**page 3, line 12**):

"Our starting point and main assumption is that the particle near-spherical-shape can be highly depolarizing, as shown in the work of Mishchenko and Hovenier (1995) and Bi et al. (2018). "

*4. Figure 2: The images seem to be incorrectly placed, given the aspect ratios. Oblate sphere should be flattened and prolate stretched.*

   **REPLY:** Thank you for noticing this, the figure has been updated.

*5. Page 6, Line 20 - 26 and Figure 6: The latitude limits in the text do not match those in the Figure and the red dashed lines do not match the section highlighted in Figure 4. But even more importantly, the corresponding browse images indicate a complex mixture of aerosol and ice from cirrus clouds which would explain the high depolarization ratios in these cases.*

*https://www.calipso.larc.nasa.gov/products/lidar/browse_images/show_detail.php?s=production&v=V4-10&browse_date=2017-08-15&orbit_time=17-55-33&page=3&granule_name=CAL_LID_L1-Standard-V4-10.2017-08-15T17-55-33ZD.hdf*

*https://wwwcalipso.larc.nasa.gov/products/lidar/browse_images/show_detail.php?s=production&v=V4-10&browse_date=2017-08-15&orbit_time=17-55-33&page=4&granule_name=CAL_LID_L1-Standard-V4-10.2017-08-15T17-55-33ZD.hdf*

**REPLY:** Figure 6 has been corrected. Regarding the next part of the question, CALIOP measurements at the northeastern Canada on 15 August 2017 (https://www-calipso.larc.nasa.gov/products/lidar/browse_images/show_v4_detail.php?s=production&v=V4-10&browse_date=2017-08-15&orbit_time=17-55-33&page=3&granule_name=CAL_LID_L1-Standard-V4-10.2017-08-15T17-55-33ZD.hdf), show the stratospheric smoke layer at 11-14.5 km, where radiosonde measurements show temperatures below -40°C (Fig. 1 below). The radiosonde temperature profiles are from three stations close to the position of the smoke plume (Fig. 4b in manuscript): Churchill (Lat: 58.73, Lon: -94.08), Inukjuak (Lat: 58.45, Lon: -78.11) and Baker Lake (Lat: 64.31, Lon: -96.00).  Moreover, the ground-based lidar measurements on 23 August 2017 at Leipzig, show the stratospheric smoke layer at 14-16 km, where radiosonde measurements from the closest station (Lindenberg) provide temperatures below -50°C. Indeed, at such low temperatures homogenous ice formation can occur (Wallace and Hobbs, 2006; Fig. 6.29). However**,** the CALIOP PLDR values are below 20% both for the aforementioned overpass and for the closest overpass from Leipzig on 23 August 2017 (https://www-calipso.larc.nasa.gov/products/lidar/browse_images/show_v4_detail.php?s=production&v=V4-

10&browse_date=2017-08-23&orbit_time=01-29-01&page=1&granule_name=CAL_LID_L1-Standard-V4-10.2017-08-23T01-29-01ZN.hdf(~90 km away and approximately 1 hour after the end of the ground based lidar measurements reported from Haarig et al., 2018), while the attenuated color ratio (i.e., the ratio of particle backscatter coefficient at 532nm to particle backscatter coefficient at 1064nm) is below 1. Further analysis of CALIOP data provides a mean (median) value of the backscatter related Angstrom exponent at 532/1064nm of 0.9 (0.9) with a standard deviation on 1.07. For PLDR, typical values for cirrus clouds are usually no less than 40% (Chen at al., 2002; Noel et al., 2002; Voudouri et al., 2020) and the color ratio is expected to be close to 1 due to the large size of ice crystals compared to the lidar wavelengths. For the Angstrom exponent values close to zero are expected, although, as indicated by the large standard deviation, CALIPSO data are highly noisy at these altitudes

Moreover, for the overflight close to Leipzig on 23 August 2017, the lidar ratio (LR) measured from the ground-based system is (66 ± 12) sr at 532nm. This is similar to the LR observed in the past for aged smoke particles (i.e. Fiebig et al., 2002; Veselovskii et al., 2015; Burton et al., 2012) but quite high for cirrus clouds (Gouveia et al., 2017).

Based on the above, although we cannot exclude the possibility of small ice crystals formed inside the stratospheric plume, we believe that the aforementioned characteristics indicate that this is not an ice cloud but rather a large smoke plume.

[Figure]

**Figure 1.** Corrected surface reflectance from MODIS on 15 August 2017, over-plotted with the PyroCb aerosol index product from Suomi NPP/OMPS (in yellow). Green stars indicate the position of the radiosonde stations used, while the green line marks the CALIPSO overflight during 18:22 – 18:35 UTC. The map is generated from the NASA Worldview Snapshots.

[Figure]

**Figure 2.** Same as Fig. 1 but for 23 August 2017. The CALIPSO overflight is approximately 90 km from Leipzig station, at 01:23 – 01:48 UTC, 1 hour after the end of the ground based lidar measurements.

[Figure]

**Figure 3.** Radiosonde temperature (T) profiles from Churchill (Ch), Inukjuak (In), Baker Lake (Bl) and Lindenberg stations. Solid lines denote the measurements at 00:00 UTC, while dashed lines the measurements at 12:00 UTC. For the first three stations (Ch, In, Bl) measurements from 15 August 2017 are used, while for Li station from 23 August 2017. The pink box indicates the height of the smoke plume above northeastern Canada (11 -14 km) and the blue box the height of the plume after 8 days above Leipzig station (15 - 16 km).

To highlight this comment, we included the following paragraph in the revised manuscript: **(page 6, line 24)**: "Owning to the altitude of the smoke plume, one could attribute such PLDR values to the beginning of ice formation. Indeed, radiosonde temperature profiles from three stations located underneath the smoke plume (green stars in Fig.3b), reveal that the temperature above 11 km drops below -40C, at which point homogeneous ice formation can occur (Wallace and Hobbs, 2006). However, the PLDR values of cirrus clouds are usually no less than 40% (Chen at al., 2002; Noel et al., 2002; Voudouri et al., 2020). Further analysis of CALIOP data provides a mean (median) value of the backscatter related Angstrom exponent (BAE) at 532/1064nm of 0.9 (0.9) with a standard deviation on 1.07. For the BAE values close to zero are expected for cirrus clouds, although, as indicated by the large standard deviation, CALIPSO data are highly noisy at these

altitudes. A recent study by Yu et al. (2019) also showed that the largest fraction of stratospheric smoke particles consisted of organic carbon (98% compared to 2% for black carbon). Particles of such high organic carbon content serve poorly as ice nuclei (Kanji et al., 2017; Phillips et al., 2013). Although the possibility of small ice crystals formed inside the smoke layers cannot be excluded, (largely due to the absence of in situ measurements) the aforementioned characteristics indicate that this plume consists of smoke particles rather than ice crystals."

**References** *(that are not included in the initial version of the manuscript)*:

Burton, S. P., Ferrare, R. A., Hostetler, C. A., Hair, J. W., Rogers, R. R., Obland, M. D., Butler, C. F., Cook, A. L., Harper, D. B., and Froyd, K. D.: Aerosol classification using airborne High Spectral Resolution Lidar measurements – methodology and examples, Atmos. Meas. Tech., 5, 73–98, https://doi.org/10.5194/amt-5-73- 2012, 2012.

Chen WN, Chiang CW, Nee JB. Lidar ratio and depolarization ratio for cirrus clouds. *Appl Opt*. 2002;41(30):6470-6476. doi:10.1364/ao.41.006470

Fiebig, M., Petzold, A., Wandinger, U., Wendisch, M., Kiemle, C., Stifter, A., Ebert, M., Rother, T., and Leiterer, U.: Optical closure for an aerosol column: Method, accuracy, and inferable properties applied to a biomass-burning aerosol and its radiative forcing, J. Geophys. Res., 107, 8130, https://doi.org/10.1029/2000JD000192, 2002.

Gouveia, D. A., Barja, B., Barbosa, H. M. J., Seifert, P., Baars, H., Pauliquevis, T., and Artaxo, P.: Optical and geometrical properties of cirrus clouds in Amazonia derived from 1 year of ground-based lidar measurements, Atmos. Chem. Phys., 17, 3619–3636, https://doi.org/10.5194/acp-17-3619-2017, 2017.

Kanji, Z. A., Ladino, L. A., Wex, H., Boose, Y., Burkert-Kohn, M., Cziczo, D. J., and Krämer, M.: Chapter 1: Overview of ice nucleating particles, Meteor Monogr., Am. Meteorol. Soc., 58, 1.1-1.33, https://doi.org/10.1175/amsmonographs-d-16-0006.1, 2017.

Noel, V., Chepfer, H., Ledanois, G., Delaval, A., and Flamant, P.: Classification of Particle Effective Shape Ratios in Cirrus Clouds Based on the Lidar Depolarization Ratio, Appl. Optics, 41, 4245–4257, https://doi.org/10.1364/AO.41.004245, 2002.

Phillips, V. T. J., P. J.Demott, C.Andronache, K. A.Pratt, K. A.Prather, R.Subramanian, and C.Twohy, 2013: Improvements to an empirical parameterization of heterogeneous ice nucleation and its comparison with observations. J. Atmos. Sci., 70, 378–409, doi:https://doi.org/10.1175/JAS-D-12-080.1.

Veselovskii, I., Whiteman, D. N., Korenskiy, M., Suvorina, A., Kolgotin, A., Lyapustin, A., Wang, Y., Chin, M., Bian, H., Kucsera, T. L., Pérez-Ramírez, D., and Holben, B.: Characterization of forest fire smoke event near Washington, DC in summer 2013 with multi-wavelength lidar, Atmos. Chem. Phys., 15, 1647– 1660, https://doi.org/10.5194/acp-15-1647-2015, 2015.

Voudouri, K. A., Giannakaki, E., Komppula, M., and Balis, D.: Variability in cirrus cloud properties using a Polly$^{XT}$ Raman lidar over high and tropical latitudes, Atmos. Chem. Phys., 20, 4427–4444, https://doi.org/10.5194/acp-20-4427-2020, 2020.

Wallace, J.M., Hobbs, P.V. Atmospheric Science: An Introductory Survey: Second Edition (2006), DOI: 10.1016/C2009-0-00034-8

*6. The images from Haarig et al 2018 are more convincing for the case the authors are making. Perhaps another CALIPSO image should be used if the authors would like to show satellite measurements.*

**REPLY:** We improved the images to better demonstrate our arguments.

*7. Page 8, Line 12-13: The authors state: What is interesting here is that the retrieved sizes for near-spherical smoke particles are absent in the AERONET climatology product. However, the difference in using a mono-modal versus bi-modal distribution is something that should be explored or at least discussed more in the text. Are we seeing the result of an "effective" radius in a mon-modal distribution that presents a possible solution for the high depolarization and spectral dependence, when there could also be another solution obtained from a mixture in a bi-modal distribution?*

**REPLY:** Thank you for this comment. We have provided a reply to a similar comment made by anonymous Reviewer 1: Regarding the kind of the distribution used, as discussed in Hansen and Travis (1974), the sensitivity of the optical properties of the particles to the type of the size distribution is limited. Regarding the second (coarse) mode, similar simulations presented by Bi et al., (2018; Fig 2), suggest that for near-spherical particles the measured spectral dependence of PLDR could not be reproduced by coarse mode particles. Thus,

an optically significant coarse mode would have to be investigated with a different shape model. Maybe the reviewer is further interested in the answer provided for Comment 11 made by anonymous Reviewer 3.

In the revised version of the manuscript, we have also included the following paragraph (**page 4, line 20**):

"The fixed width of the size distribution $\sigma_g$ is again a simplification we used in order to reduce the retrieval complexity, considering that this parameter does not greatly affect the lidar-derived optical properties (e.g. Burton et al., 2016). Choosing a log-normal size distribution over any other plausible type of distribution is not expected to alter our results significantly (Hansen and Travis, 1974)."

**References** *(that are not included in the initial version of the manuscript)*:

Hansen, J.E., Travis, L.D. Light scattering in planetary atmospheres. Space Sci Rev 16, 527–610 (1974). https://doi.org/10.1007/BF00168069

*8. Sections 4.2 and 4.3: Is this all that can be said for these solutions... we tried these shapes and they didn't work? It would seem that more could be gleaned from these failed attempts. For instance, these shapes show the correct trend in spectral dependence even though the absolute values/relative differences do not fit the measurements.*

**REPLY:** We would like to thank the reviewer for this comment. Updated discussion on the results for Chebyshev particles has been included in the manuscript *page 5, line 4*):

"For Chebyshev particles of second ($T_2$) and fourth degree ($T_4$) used herein, the search in the constructed look-up-tables provided the solutions listed in Table 4. For all the solutions, deformation parameter for Chebyshev particles of the second degree ranges from $u = -0.25$ to 0.15, while for particles of the fourth degree only one solution was found with $u = -0.1$. These $u$ values suggest small deviations from sphericity, meaning that these morphologies also resemble near-spherical shapes. Only for two cases the size of the particles was found to be larger than for the near-spherical shaped particles. In particular $r_g$ ranges from 0.15μm ($reff = 0.2$μm) to 0.55μm ($reff = 0.8$μm). For the complex refractive index, values in some cases exceed the corresponding values for near-spherical particles. The imaginary part $m_i$ ranges from i0.005 to i0.055, and the real part $m_r$ ranges from 1.35 to 1.8. The minimization of the cost function (Eq. 8) is achieved for Chebyshev particles of the second degree with u = -0.25 (resembling an oblate near-spherical particle), complex refractive index $m = 1.65 + i(0.03)$ and mean geometric radius $r_g = 0.2$μm. For Chebyshev particles of the fourth degree, the

sole solution presented values $u$ = -0.1, $m$ = 1.35 + i(0.01) and $r_g$ = 0.55µm. All possible solutions as well as those that minimize the cost function are presented in Fig. 10 and 11."

We decided to remove fractal aggregates from the present study, since as pointed out by Anonymous Reviewer 3 the range of the parameters used in our study to model fractal aggregates was limited.

*9. Page 9, Line 18: Can the authors offer an explanation for the low angstrom exponents other than coarse particles?*

**REPLY:** We have provided an answer to a similar comment made by anonymous Reviewer 3. We summarize also here: According to Eck et al. (1999), the strong curvature between the extinction related Angstrom exponent (EAE) at 355/532nm (-0.3 ± 0.4) and the corresponding values at 532/1064nm (0.85 ± 0.3) can be attributed to the pronounced accumulation mode of the size distribution, which is in good agreement with the retrieved size distribution for near-spherical particles of $reff$ = 0.38µm. Another possible reason could be a spectrally-dependent absorption, although this is not shown in our results due to the assumed spectrally-independent value of the imaginary part of refractive index. To address this comment we added the following paragraph to the manuscript (**page 8, line 22**):

"We note here that all the retrievals indicate fine particles, with mean geometric radius that does not exceed the value of 0.35µm. The simulations presented by Bi et al., (2018; Fig. 2) suggest that for the near-spherical particles the measured spectral dependence of PLDR (steeply decreasing from the UV to the Near-IR) could not be reproduced by coarse particles. Thus, the possibility of an optically significant coarse mode would have to be investigated with a different shape model. In any case though, the retrieved fine mode is in good agreement with in-situ measurements of aged smoke particles (i.e. Dahlkoetter et al., 2014). The presence of a pronounced accumulation mode is also suggested by the extinction related Angstrom exponent (EAE) measured in Leipzig (-0.3±0.4 at 355/532nm and 0.85 0.3 at 532/1064nm). According to Eck et al. (1999), a strong spectral slope in EAE can be associated with a prominent accumulation mode of the size distribution for smoke particles"

**References** *(that are not included in the initial version of the manuscript)*:

Eck, T. F., Holben, B. N., Reid, J. S., Dubovik, O., Smirnov, A., O'Neill, N. T., Slutsker, I., and Kinne, S.:

Wavelength dependence of the optical depth of biomass burning, urban, and desert dust aerosols, Journal of

Geophysical Research: Atmospheres, 104, 31333-31349, 10.1029/1999JD900923, 1999.

---

## Author Comment (AC3) · 16 Jul 2020

*The manuscript topic fits well within the journal scope is providing new insights on biomass burning aerosol layers. Nevertheless, it needs major revisions before being ready for publication.*

**REPLY:** Thanks for your helpful comments. Corrections have been made considering your suggestions as well as other reviewers'. Please find our point-by-point response and first revised version in the supplement. Remark: The figure numbers and the page numbers in the referee comments and in our replies correspond to the original manuscript.

*Major comments:*

*1. A substantiated and consolidated verification of the measurement quality and the potential role of systematic errors affecting the measurements is a preliminary paramount step when such high PLDR values are measured. This is particularly true for stratospheric aerosols as calibration of aerosol depolarization measurements of stratospheric particles is quite difficult and cannot rely on molecular calibration approach.*

**REPLY:** Thank you for this comment. Indeed, the calibration of the depolarization measurements is very crucial for any aerosol study. For the calibration of the depolarization measurements used in this study we followed the "*Δ±45 depolarization calibration*" method proposed by Freudenthaler et al. (2009). Specifically, for the PLDR measurements used here, the systematic errors are 0.015 at 355nm, 0.006 at 532nm and 0.007 at 1064nm as presented in Haarig et al. (2018). A detailed discussion on the parameters affecting the depolarization measurements of the BERTHA lidar system is presented in Haarig et al. (2017) (APPENDIX A).

To highlight this comment, we added the following paragraph to the manuscript (**page 7, line 14**):

"To ensure the high quality of depolarization measurements, the $\Delta\pm45$ depolarization calibration method proposed by Freudenthaler et al. (2009) was followed, while the effect of different parameters on the depolarization measurements of the BERTHA lidar system has been carefully assessed and is presented in detail in Haarig et al. (2017)."

**References** *(that are not included in the initial version of the manuscript)*:

Freudenthaler, V., Esselborn, M., Wiegner, M., Heese, B., Tesche, M., Ansmann, A., Müller, D., Althausen, D., Wirth, M., Fix, A., Ehret, G., Knippertz, P., Toledano, C., Gasteiger, J., Garhammer, M., Seefeldner, M.: Depolarization ratio profiling at several wavelengths in pure Saharan dust during SAMUM 2006, Tellus B: Chemical and Physical Meteorology, 61:1, 165-179, DOI: 10.1111/j.1600-0889.2008.00396.x, 2009.

Haarig, M., Ansmann, A., Althausen, D., Klepel, A., Groß, S., Freudenthaler, V., Toledano, C., Mamouri, R.-E., Farrell, D. A., Prescod, D. A., Marinou, E., Burton, S. P., Gasteiger, J., Engelmann, R., and Baars, H.: Triple-wavelength depolarization-ratio profiling of Saharan dust over Barbados during SALTRACE in 2013 and 2014, Atmos. Chem. Phys., 17, 10767–10794, https://doi.org/10.5194/acp-17-10767-2017, 2017.

*2.The fact that such high PLDR values were reproduced using T-matrix simulations, assuming near-spherical shapes, for biomass burning is not itself a verification of the fact that observed particles were indeed transported stratospheric smoke plumes. More information on possible particle composition, and its possible organic origin, should be inferred from other optical measurements (multi-wavelength particle extinction and backscatter measurements).*

**REPLY:** Thank you very much for this comment. Indeed, the origin/composition of the particles cannot be deduced only from the measurements presented in the manuscript (multi-wavelength PLDR and LR measurements). Detailed discussion on the transport of the smoke plumes that are presented in our analysis is included in several previous studies referring to the Canadian wildfires of August 2017. For example, Khaykin et al. (2018) present CALIPSO data that are used to follow the evolution of the plume since two days after the PyroCb eruption on 14 August 2017 (Peterson et al., 2017) to 30 August 2017 (see Fig. 3a in supplement S2 from Khaykin et al., 2018). The ground-based lidar observations at Leipzig on 23 August 2017 presented in the manuscript, observe the smoke plume, which was located above Germany during 21 – 24 August 2017 (Khaykin et al., 2018). In Ansmann et al. (2018), HYSPLIT backward and forward trajectories were used to depict the route of the smoke plume from North America to central Europe and identify the smoke source regions. Results were found to be in good agreement with CALIPSO observations and UV aerosol index maps from OMPS presented in Khaykin et al. (2018). In Hu et al. (2019) MODIS maps, UV aerosol index from OMPS as the CO product from AIRS were used to determine whether the observed aerosol plumes over northern France were indeed smoke transported from Canada. Indeed, the

strong spatio-temporal correlation between UV aerosol index and CO revealed the smoke presence. Apart from the high PLDR values measured from the ground-based lidar system in Leipzig, lidar ratio (LR) values are also available at 3 wavelengths and used in our simulations: $40 \pm 16sr$, $66 \pm 12sr$, $92 \pm 27sr$ at 355, 532 and 1064nm. Although LR of smoke presents a large variability due to different particle characteristics between fresh and aged smoke particles, these LR values are in good agreement with past measurements for smoke LR at 355 and 532nm (i.e. Fiebig et al., 2002; Muller et al., 2005; Ortiz-Amezua et al., 2017).

**References** *(that are not included in the initial version of the manuscript)*:

Fiebig, M., Petzold, A., Wandinger, U., Wendisch, M., Kiemle, C., Stifter, A., Ebert, M., Rother, T., and Leiterer, U.: Optical closure for an aerosol column: Method, accuracy, and inferable properties applied to a biomass-burning aerosol and its radiative forcing, J. Geophys. Res., 107, 8130, https://doi.org/10.1029/2000JD000192, 2002.

Ortiz-Amezcua, P., Guerrero-Rascado, J. L., Granados-Muñoz, M. J., Benavent-Oltra, J. A., Böckmann, C., Samaras, S., Stachlewska, I. S., Janicka, L., Baars, H., Bohlmann, S., and AladosArboledas, L.: Microphysical characterization of long-range transported biomass burning particles from North America at three EARLINET stations, Atmos. Chem. Phys., 17, 5931–5946, https://doi.org/10.5194/acp-17-5931-2017, 2017.

*3. Because of 2) the proposed approach is rater weak. It is not possible to generalize statements just from a single case study. Moreover, it seems a sort of ill-posed problem and the minimum in Eq. 8 might be relative, i.e. what happens if instead a mono-modal distribution a bimodal is chosen? or a gamma instead of normal distribution? Probably Eq. 8 will provide independently a solution.*

**REPLY:** As discussed in Hansen and Travis (1974), the sensitivity of the optical properties of the particles to different types of size distributions (e.g. standard gamma, log normal, bimodal and a power-law) is limited. Maybe the reviewer is also interested in the answer provided for a similar comment (Comment 11) made by anonymous Reviewer 3.

In the revised version of the manuscript, we have included the following (**page 4, line 20**):

"The fixed width of the size distribution $\sigma_g$ is again a simplification we used in order to reduce the retrieval complexity, considering that this parameter does not greatly affect the lidar-derived optical properties (e.g.

Burton et al., 2016). Choosing a log-normal size distribution over any other plausible type of distribution is not expected to alter our results significantly (Hansen and Travis, 1974)."

Regarding the first part of the comment, we agree with the reviewer. For this reason, we updated the manuscript, including the retrievals for all available measurements of stratospheric smoke in the literature, using the proposed near-spherical model. Figure 1 below presents some examples of successful reproduction of the measurements for all the cases assuming near-spherical shapes, and Table 2 below presents the retrieved values for the mean axial ratio $\varepsilon_s$ of the near-spherical shapes, the complex refractive index $m$ and the effective radius $reff$ of the particles. All the retrievals (using near-spherical and Chebyshev particles) are available in the manuscript Supplement (for Hu et al., 2019 fitting of the measurements of 31 August 2017 are presented. For Ohneiser et al., 2020 fitting of the measurements of 8 January 2020 are presented).

Furthermore, we added the following section to the text (**page 8, line 23**):

"Although the available literature on the PLDR and LR values of stratospheric smoke is for now limited, we see that we can reproduce all reported of PLDR and LR using the near-spherical shape model (Table 1-9 and Fig. 1-9 in the Supplement). All cases listed in Table 2 are associated with Pyro-cumulonimbus activity. As already mentioned the case studies of Burton et al. (2015), Hu et al. (2019) and Haarig et al. (2018) refer to Canadian smoke, while the most recent case study presented by Ohneiser et al. (2020) refer to Australian wildfires of 2019-2020. Table 5 present the retrieved mean axial ratio, complex refractive index and geometric radius of the size distribution. For Hu et al. (2019), measurements on 24, 29 and 31 August were reported. For Ohneiser et al. (2020) measurements on 8, 9 and 10 January 2020 were reported."

**Table. 1:** Reported PLDR and LR values for UTLS smoke. For Hu et al. (2019) and Ohneiser et al. (2020), one of the available observations is included in the table.

| | PLDR$_{355}$ (%) | PLDR $_{532}$ (%) | PLDR $_{1064}$ (%) | LR$_{355}$ (sr) | LR$_{532}$ (sr) | LR$_{1064}$ (sr) |
|---|---|---|---|---|---|---|
| Burton et al. (2015) | $20.3 \pm 3.6$ | $9.3 \pm 1.5$ | $1.8 \pm 0.2$ | X | X | X |
| Hu et al. (2019) | $24 \pm 4$ | $19 \pm 3$ | $5 \pm 1$ | $41 \pm 7$ | $54 \pm 9$ | X |

| Ohneiser et al. (2020) | 26 ± 5.2 | 15 ± 1.5 | X | 53 ± 15.9 | 76 ± 15.2 | X |

[Figure]

**Figure 1.** Example fittings of the PLDR and LR measurements presented in Hu et al. (2019), Burton et al. (2015) and Ohneiser at al. (2020), using the near-spherical model. First two cases refer to Canadian wildfires of 2017 and 2014, respectively. The third case refers to the Australian wildfires of last 2019 – 2020. All cases are associated with PyroCb activity. TM in the legend stands for the T-matrix simulations with near-spherical particles: blue circles denote to the simulations reproducing the observations of Hu (2019), pink circles denote the simulations reproducing the observations of Burton (2015), and green circles denote to the simulations reproducing the observations of Ohneiser (2020). All of the retrievals are included in the manuscript Supplement.

**Table 2.** The simulations with the near-spherical shape model, used to reproduce the measurements presented in Table 1.

|  | $r_g$ (µm) | $\varepsilon_s$ | $m_i$ | $m_r$ |
|---|---|---|---|---|
| Burton et al. (2015) | 0.3 | 1.15 | 0.005 | 1.45 |
| Hu et al. (2019) | 0.25 | 1.45 | 0.02 | 1.55 |
| Ohneiser et al. (2020) | 0.35 | 0.9 | 0.035 | 1.45 |

**References** *(that are not included in the initial version of the manuscript)*:

Hansen, J.E., Travis, L.D. Light scattering in planetary atmospheres. Space Sci Rev 16, 527–610 (1974). https://doi.org/10.1007/BF00168069

Ohneiser, K., Ansmann, A., Baars, H., Seifert, P., Barja, B., Jimenez, C., Radenz, M., Teisseire, A., Floutsi, A., Haarig, M., Foth, A., Chudnovsky, A., Engelmann, R., Zamorano, F., Bühl, J., and Wandinger, U.: Smoke of extreme Australian bushfires observed in the stratosphere over Punta Arenas, Chile, in January 2020: optical thickness, lidar ratios, and depolarization ratios at 355 and 532 nm, Atmos. Chem. Phys., 20, 8003–8015, https://doi.org/10.5194/acp-20-8003-2020, 2020.

*4. As stated by Sassen and Khvorostyanov, smoke can directly act as ice nuclei before liquid clouds form (https://iopscience.iop.org/article/10.1088/1748-9326/3/2/025006). This fact can partially explain the higher PLDR (considering a process in progress). This aspect, very likely is not mentioned in the manuscript and can be the reason of PLDR increase.*

**REPLY:** We agree with the reviewer that the PLDR values alone could indicate the formation of ice crystals inside the stratospheric smoke layer. However, the reported PLDR values of ~20% at 532nm are small compared to those usually observed (>40%) for cirrus clouds containing ice crystals (Chen at al., 2002; Noel et al., 2002; Voudouri et al., 2020). Furthermore, the available data from Leipzig include also the lidar ratio (LR) values of 66 ± 12 sr at 532nm. This is similar to the LR observed in the past for aged smoke particles (i.e. Fiebig et al., 2002; Veselovskii et al., 2015; Burton et al., 2012) but quite high for cirrus clouds which present values of the order of 25 sr (Gouveia et al., 2017). A recent study by Yu et al. (2019) also showed that the largest fraction of stratospheric smoke particles consisted of organic carbon (98% compared to 2% for black carbon). Particles of such high organic carbon content serve poorly as ice nuclei (Kanji et al., 2017; Phillips et al., 2013).

We would also like to refer the reviewer to Comment 5 from anonymous Reviewer 3, who raised a similar concern on ice formation.

To highlight this for the reader we included the following in the manuscript **(page 6, line 24):**

"Owning to the altitude of the smoke plume, one could attribute such PLDR values to the beginning of ice formation. Indeed, radiosonde temperature profiles from three stations located underneath the smoke plume

(green stars in Fig.3b), reveal that the temperature above 11 km drops below -40C, at which point homogeneous ice formation can occur (Wallace and Hobbs, 2006). However, the PLDR values of cirrus clouds are usually no less than 40% (Chen at al., 2002; Noel et al., 2002; Voudouri et al., 2020). A recent study by Yu et al. (2019) also showed that the largest fraction of stratospheric smoke particles consisted of organic carbon (98% compared to 2% for black carbon). Particles of such high organic carbon content serve poorly as ice nuclei (Kanji et al., 2017; Phillips et al., 2013). Although the possibility of small ice crystals formed inside the smoke layers cannot be excluded, (largely due to the absence of in situ measurements) the aforementioned characteristics indicate that this plume consists of smoke particles rather than ice crystals."

**References** *(that are not included in the initial version of the manuscript)*:

Burton, S. P., Ferrare, R. A., Hostetler, C. A., Hair, J. W., Rogers, R. R., Obland, M. D., Butler, C. F., Cook, A. L., Harper, D. B., and Froyd, K. D.: Aerosol classification using airborne High Spectral Resolution Lidar measurements – methodology and examples, Atmos. Meas. Tech., 5, 73–98, https://doi.org/10.5194/amt-5-73- 2012, 2012.

Chen WN, Chiang CW, Nee JB. Lidar ratio and depolarization ratio for cirrus clouds. *Appl Opt*. 2002;41(30):6470-6476. doi:10.1364/ao.41.006470

Fiebig, M., Petzold, A., Wandinger, U., Wendisch, M., Kiemle, C., Stifter, A., Ebert, M., Rother, T., and Leiterer, U.: Optical closure for an aerosol column: Method, accuracy, and inferable properties applied to a biomass-burning aerosol and its radiative forcing, J. Geophys. Res., 107, 8130, https://doi.org/10.1029/2000JD000192, 2002.

Gouveia, D. A., Barja, B., Barbosa, H. M. J., Seifert, P., Baars, H., Pauliquevis, T., and Artaxo, P.: Optical and geometrical properties of cirrus clouds in Amazonia derived from 1 year of ground-based lidar measurements, Atmos. Chem. Phys., 17, 3619–3636, https://doi.org/10.5194/acp-17-3619-2017, 2017.

Kanji, Z. A., Ladino, L. A., Wex, H., Boose, Y., Burkert-Kohn, M., Cziczo, D. J., and Krämer, M.: Chapter 1: Overview of ice nucleating particles, Meteor Monogr., Am. Meteorol. Soc., 58, 1.1-1.33, https://doi.org/10.1175/amsmonographs-d-16-0006.1, 2017.

Noel, V., Chepfer, H., Ledanois, G., Delaval, A., and Flamant, P.: Classification of Particle Effective Shape Ratios in Cirrus Clouds Based on the Lidar Depolarization Ratio, Appl. Optics, 41, 4245–4257, https://doi.org/10.1364/AO.41.004245, 2002.

Phillips, V. T. J., P. J.Demott, C.Andronache, K. A.Pratt, K. A.Prather, R.Subramanian, and C.Twohy, 2013: Improvements to an empirical parameterization of heterogeneous ice nucleation and its comparison with observations. J. Atmos. Sci., 70, 378–409, doi:https://doi.org/10.1175/JAS-D-12-080.1.

Veselovskii, I., Whiteman, D. N., Korenskiy, M., Suvorina, A., Kolgotin, A., Lyapustin, A., Wang, Y., Chin, M., Bian, H., Kucsera, T. L., Pérez-Ramírez, D., and Holben, B.: Characterization of forest fire smoke event near Washington, DC in summer 2013 with multi-wavelength lidar, Atmos. Chem. Phys., 15, 1647–1660, https://doi.org/10.5194/acp-15-1647-2015, 2015.

Voudouri, K. A., Giannakaki, E., Komppula, M., and Balis, D.: Variability in cirrus cloud properties using a Polly$^{XT}$ Raman lidar over high and tropical latitudes, Atmos. Chem. Phys., 20, 4427–4444, https://doi.org/10.5194/acp-20-4427-2020, 2020.

*5) The simulations themselves are not original as in fact similar simulations were performed in the past by Bi et al. 2018, Mishchenko et al. 2016; Ishimoto et al., 2019, as the authors explicitly admit. What is different with respect to those manuscript?*

**REPLY:** Bi et al. (2018) is an interesting modeling study on the properties of spheroid and super-ellipsoid particles for a large suite of refractive indices and size parameters. It is though a generic study, not focused on stratospheric smoke particles. Also, the simulations in Bi et al. (2018) refer only to PLDR and not to other intensive properties (e.g. LR) as we do in our study.  On the other hand, Mishchenko et al. (2016) used four different models to reproduce the PLDR values observed by Burton et al., (2015). Our results are comparable, but the study is only limited to PLDR since there were no available LR measurements at the time. Ishimoto et al. (2019) use fractal aggregates coated by water soluble materials. In this study both the PLDR and LR are examined, but the simulations refer only to monodisperse particles. The results are comparable to ours only for coated fractals, producing a shape that closely resembles the near-spherical shape (i.e. shapes of "Type-B, size 11, Vr = 20" shown in Fig. 4 of Ishimoto et al. 2019).

In our study we propose a simple model of compact near-spherical particles, that can reproduce both the PLDR and LR values measured by sophisticated lidar systems, part of the EARLINET, that are capable of providing quality-assured retrievals for stratospheric smoke particles. We further examine whether this model could be used on an operational level to extend the AERONET retrieval scheme. The introduction of

the manuscript has been updated in order to present how our research is differentiated by previous research (**page 3, line 9**):

"In contrast to prior studies, for our investigation for the stratospheric smoke originating from the Canadian wildfires, we do not adopt morphologically complex shapes of bare or coated smoke aggregates, which are associated with excessive computations. Instead, we propose a much simpler model of compact near-spherical particles. Our starting point and main assumption is that the particle near-spherical-shape can be highly depolarizing, as shown in the work of Mishchenko and Hovenier (1995) and Bi et al. (2018). Our analysis shows that for the Canadian stratospheric smoke observed above Europe in August 2017, the PLDR and LR measurements along with their spectral dependence, can be successfully reproduced with the proposed model of compact near-spherical particles. The size and refractive index of the particles are estimated as well, and seem to agree well with past observations for aged smoke. We further examine the capability of this model to be used on an operational level and in particular as an extension to the AERONET operational aerosol retrieval (Dubovik et al., 2006), since it provides a much simpler and faster solution with respect to more complicated shapes for stratospheric smoke particles (e.g. Mishchenko et al.,2016; Ishimoto et al., 2019)."

*6) The title (i found it funny) might be misinterpreted and considered inappropriate.*
**REPLY:** Thank you for your comment.

We reply to specific comments in the attached manuscript below:

***Page 1, line 15:*** *We added: "of axial ratio 0.7 to 1.5"*

***Page 2, line 1:*** We rephrased to: "Smoke particles in the atmosphere can be identified with lidar measurements which provide valuable information on the optical properties of aerosol particles, such as the depolarization of the backscattered light in terms of the particle linear depolarization ratio (PLDR)."

***Page 9, line 4:*** please see response to Comment 3.

---

## Author Comment (AC4) · 16 Jul 2020

*In this note I offer comments limited to Section 3 and 5 regarding the evidence for mixing of volcanic sulfuric acid with smoke, for the authors to consider. This manuscript makes an effort to suggest possible effects of mixing volcanic H2SO4 with smoke and thereby creating a "new black" particle linear depolarization ratio (PLDR). The example given is a volcanic SO2 plume on 8 August 2017. In looking at the figures and text I see some apparent inconsistencies and possible misinterpretation.*

**REPLY:** We thank very much Dr. Fromm for his careful reading, comments and suggestions, which we address in the following. We agree that at the moment the discussion on the possible physical explanation for the increased smoke PLDR is not complete. Thus we decided to remove this section from the revised manuscript. More efforts will be made to investigate this issue and re-assess the possible coexistence of smoke particles and particles of sulfuring nature and whether this could affect the former in such a way to form near-spherical particles.

*Figure 15 shows a map that includes upper tropospheric SO2, and attributes that to Shiveluch. (The bottom panel of Figure 15 is confusing in that it refers to Himawari-9 imagery but the analysis illustrated is something else.) Although an eruption on 8 August at Shiveluch was reported, it seems much more likely that the SO2 came from a 7 August 2017 eruption of Bogoslof in the Aleutians. https://www.adn.com/alaska-news/2017/08/07/alaskas-tiny-bogoslof-volcano-eruptsagain-sending-an-ash-cloud-miles-above-the-aleutians/ https://volcano.si.edu/volcano.cfm?vn=311300. It becomes apparent that the SO2 derives from this eruption when one follows the OMPS SO2 back in time. The plume starts on 7 Aug right near Bogoslof (credit NASA Worldview). Regardless of the source of the SO2, the only evidence suggestive of volcanic influence is the map of 8 August SO2 (Figure 15). The pyroconvection in Canada leading to the stratospheric smoke occurred 4 days later according to papers the authors cite. Presumably if volcanic sulfur was responsible for the months of double-digit PLDR "new black" one might expect to see a robust volcanic signature close to British Columba on or much closer to 12 August. What can be shown in that regard?*

**REPLY:** Thank you for bringing this information to our attention. Indeed, the two eruptions occurred one day apart. To determine whether the plume (which at the time we believed came from Shiveluch) was transported towards the area of British Columbia, we performed forward trajectory and dispersion analysis using the offline

coupled atmospheric and dispersion model FLEXPART-WRF (Brioude et al., 2013). Updated simulations with the FLEXPART showed that both plumes (from Shiveluch and Bogoslof) travelled towards the same direction, to the area of British Columbia. Nevertheless, we have deleted this part from our paper, for the reason mentioned above.

*The paper refers to the 8 August CALIPSO measurement as "daytime" when it is in fact a night-time orbit segment. Consequently, the connection made between this CALIPSO measurement and the daytime 8 August MODIS image in Figure 4 is inaccurate. On a technical note, the red dashed lines in Figure 6 for the 15 August CALIPSO data are not where the text directs the reader: the 13 km smoke layer.*

**REPLY:** Thank you for noticing this, we have corrected the figures.

*The authors refer to a 12 August CALIPSO measurement (Page 6, line 23) but don't show any such measurement. It is apparent they meant 15 August but this needs to be clarified (if indeed they intend to show a 12 August measurement) or corrected.*

**REPLY:** Thank you for noticing this, we have corrected it.

*On page 6, line 16 the authors seem to state that between 8 and 15 August the stratospheric smoke plume had already blown to Europe: ". . .8 and 15 August 2017, when the smoke plume has already reached Europe" They do not present any data to support that and I believe there is no support for that claim. The leading edge of the plume on 15 August was still entirely over Canada.*

**REPLY:** Thank you for this comment. Indeed, the plume was evident above Europe only after late August. We corrected this sentence in the manuscript.

*In summary, I was confused by the material in Section 3 and 5 and thus was left unconvinced of any meaningful mingling of volcanic sulfates and pyroCb-injected stratospheric smoke. Presumably, if the transport pathway was what the authors claimâ˘Tpy- ˘ roconvectionâ˘Tthe sulfates would have to have been in large concentration in the ˘vicinity of the pyroCbs and in the inflow part of the atmosphere (i.e. lower troposphere). If the two mingled by virtue of UTLS sulfates in high concentration encountering the pyroCb outflow, one*

*might expect that the sulfates would be detectible leading up to the pyroCb injection. This might be an avenue for the authors to explore because there is good CALPSO coverage of the Canadian and upstream environments on all the days between 8 and 12 August. It is becoming increasingly evident that double-digit PLDR is quite common for stratospheric smoke. In personal communication with one of the coauthors, I discussed a similar phenomenon in northern summer 2014. Here is an example of double-digit PLDR of stratospheric smoke over Scandinavia at that time (credit: http://lidar.ssec.wisc.edu/) http://hsrl.ssec.wisc.edu/by_site/18/2014/08/17/am/#MF2HSRL The Black Saturday (Australia, February 2009) pyroCb stratospheric smoke also had double-digit PLDR.*

*Here is an example of week-old smoke at that time (Credit: NASA):*

*https://www-calipso.larc.nasa.gov/products/lidar/browse_images/show_detail.php?s=production&v=V4-10&browse_date=2009-02-15&orbit_time=12-52-14&page=3&granule_name=CAL_LID_L1-   Standard-V4-10.2009-02-15T12-52-14ZN.hdf*

*It is unlikely, or at least un-established, that precursor volcanic activity occurred in these 2009 and 2014 cases. Hence it would seem that there is another common bond, albeit still unresolved, embodied in this growing record of anomalously large PLDR in dry stratospheric smoke environments.*

---

## Author Response (AR2)

**Response to Anonymous Reviewer 2**

*The authors have largely resolved my previous concerns with the manuscript, and it is suitable*

*for publication with some very minor changes.*

We thank the reviewer for his/her helpful comments. Corrections have been made considering your

suggestions. Please find our point-by-point response below and in the attached revised version of the

manuscript.

Remark: The figure numbers and the page numbers in the referee comments and in our replies correspond

to the clean version of the revised manuscript.

*Minor comments:*

*1. There are some punctuation issues which should be fixed, for example:*

*Page 2, Line 7: Double periods, ".."*

*Page 7, Line 29: missing "." after (Fig. 7b)*

**REPLY:** Thank you for noticing, we revised the whole text accordingly for such punctuation mistakes,

missing gaps, etc.

*2. Page 7, Line 20: I suggest the following change: Although the possibility of small ice crystals formed*

*inside the smoke layers cannot be excluded, (largely due to the absence of in situ measurements) the*

*aforementioned characteristics indicate that this plume consists of primarily smoke particles rather than*

*ice crystals.*

*In support of this change, I refer to Peterson et al., 2018 (referenced in the manuscript) which indicates that water ice was injected into the stratosphere and also accounts for the possibility of ice in the Methods section. From their page 2: The Cloud-Aerosol Lidar with Orthogonal Polarization (CALIOP), flown aboard NASA's polar-orbiting CALIPSO satellite,16 passed over the decaying cloud shield approximately 8h after pyroCb cessation (10:45 UTC on 13 August), confirming the injection of smoke and water ice at least 1 km into the stratosphere (not shown). By the afternoon of 14 August (19:30 UTC), ice crystal influence on the CALIOP backscatter profile within the plume had diminished, and a distinct 1.5 km deep residual layer was present over northern Canada (Fig. 3).*

**REPLY:** Thank you for your comment, we revised accordingly (**page 7, line 20-22**): "Although the possibility of small ice crystals formed inside the smoke layers cannot be excluded, (largely due to the absence of in situ measurements) the aforementioned characteristics indicate that this plume consists primarily of smoke particles rather than ice crystals."

*3. Figure 13. The real part of the refractive index should be included in a key (e.g. in the P11 plot or to the side) with the same format as it appears in the figure instead of using text.*

**REPLY:** We updated Figure 13 (and all the similar figures in the manuscript supplement) as suggested by the reviewer. We also updated Figure 14 to match the format of Figure 13 (and all the similar figures in the manuscript supplement).

**Response to Anonymous Reviewer 3**

*In my opinion, the manuscript has been significantly improved. This is an interesting study and well described, and now is an enjoyable read. Most of my concerns were addressed satisfactorily. I have a*

5 *lingering concern about the conclusion stated in the abstract that "the near spherical shape (or closely similar shapes) is the only morphology found capable of reproducing the observations", which still appears to be overstated, for the same reason as discussed in the first review (Reviewer #3). Indeed, it looks like perhaps you may have simply forgotten to update the abstract, since it is substantially the same as the first version although there are many important revisions in the paper. I suggest revising the*

10 *abstract, and to make it easy, you could take your new Conclusions section as a guide. Specifically, please remove the implication that more complicated morphologies cannot reproduce the measurements (because you didn't test coating models which most likely can reproduce them, at the cost of greater complexity) either by deleting that sentence or rewording it to be clearer what was and wasn't tested. You might want to consider highlighting the AERONET-related calculations in the abstract. Also, you might*

15 *want to highlight the newness of performing microphysical retrievals of any kind on measurements that include depolarization at 3 wavelengths.*

**REPLY:** We thank the reviewer for his/her additional comments and suggestions to improve the manuscript. We specifically revised the abstract considering your comments, as follows (**page 1, line**

20 **17**):

**Abstract.** We examine the capability of near-spherical-shaped particles to reproduce the triple wavelength Particle Linear Depolarization Ratio (PLDR) and Lidar Ratio (LR) values measured over Europe for stratospheric smoke originating from Canadian wildfires. The smoke layers were detected both in the troposphere and the stratosphere, though in the latter case the particles presented PLDR values

25 of almost 18% at 532 nm as well as a strong spectral dependence from the UV to the Near-IR. Although

recent simulation studies of rather complicated smoke particle morphologies have shown that heavily coated smoke aggregates can produce large PLDR, herein we propose a much simpler model of compact near-spherical smoke particles. This assumption allows for the reproduction of the observed intensive optical properties of stratospheric smoke, as well as their spectral dependence. We further examine

5 whether an extension of the current AERONET scattering model to include the near-spherical shapes, could be of benefit to the AERONET retrieval for stratospheric smoke cases associated with enhanced PLDR. Results of our study illustrate the fact that triple wavelength PLDR and LR lidar measurements can provide us with additional insight when it comes to particle characterization.

10 *Minor comments:*

*1. page 1, line 22, change "showing" to "and show" (to make it clear that this phrase refers to "the results resented here" not "recent findings in the literature")*

**REPLY:** Thank you, the sentence is now deleted from the updated version of the abstract.

15 *2. page 2, line 30, change "cell" to "coating"*

**REPLY:** We changed it.

*3. page 3, line 5, change "circumscription" to "circumscribing"*

**REPLY:** We changed it.

*4. page 4, line 14, I suggest inserting "more complicated Chebyshev particle shapes" to be more specific.*

**REPLY:** Thank you, we added the suggested phrase.

*5. page 5, line 15, I suggest adding "whereas the values observed in this case are mostly between ____ and ____" to the end of the sentence that points out that cloud PLDR is usually > 40%.*

**REPLY:** Thank you, we added the following sentence (**page 7, line 15**):

"…whereas the values observed in this case are mostly between 15 and 25% at 532nm, and remain so during the months of August and September following the stratospheric injection (Baars et al., 2019; Hu et al., 2019).

*6. The reference for Ohneiser et al. 2020 (called out at page 8, line 30 and elsewhere) appears to be missing in the references section.*

**REPLY:** Thank you for noticing, we added the reference and we further modified all the references to better match the ACP format.

*7. page 9, line 13, I suggest being more specific by changing the end of the sentence to be something like "ten solutions which reproduce the measurements to within the measurement uncertainty"*

**REPLY:** Thank you for this comment, we reworded as suggested (**page 9, line 16**):

"Following this methodology, for the near-spherical particles ten possible solutions were found to reproduce the measurements within the measurement uncertainty. These are listed in Table 3…"

*8. Table 2 caption, I don't think the statement "all the observations" is really correct. That is, there are other cases in literature with high PLDR dating all the way back to at least Murayama et al (2004), including some you mention in the introduction. Please reword to explain what other criteria you are thinking of. Do you mean observations that include 355 nm? Something else?*

**REPLY:** We reworded as follows (**Table 2, caption**): "Also shown are multi-wavelength observations

of PLDR and LR reported in previous studies for stratospheric or tropospheric smoke particles exhibiting high PLDR values."

*9. Figure 7 caption seems to refer to an earlier version of the figure. Please revise.*

**REPLY:** Thank you for noticing, we updated the figure and revised the caption as follows (**Figure 7, caption**): "Time–height airborne lidar observations of the PLDR at 532 nm (a) Measurements were performed over the Atlantic Ocean, between 19:00 and 21:00 UTC on 7 October 2017 by the DLR "High Altitude and Long Range Research Aircraft" (HALO) in the framework of WISE mission. The track of the aircraft is shown in (b) over-plotted on Google Earth map"

*10. Figure 9 caption. Please mention the error bars in the caption. Also please consider replacing "simulations ... for various values" with more specific information, i.e. "the solutions given in Table 3 that reproduce the measurements within the measurement uncertainty"*

**REPLY:** Thank you for your comment, we revised the caption as follows (**Figure 9, caption**): "The reproduction of the measured PLDR and LR values, considering near-spherical particles. Purple circles correspond to measurements performed on 22 August 2017, at Leipzig, Germany, while purple lines correspond to the measurement uncertainties. Blue markers denote to simulations performed with the T-matrix code, assuming near-spherical particles. Each blue triangle corresponds to a different solution found to reproduce the measurements within their uncertainties, as given in Table 3. For these solutions the mean axial ratio $\varepsilon_s$, ranges from 1.1 to 1.4, the mean geometric radius $r_g$ ranges from 0.25 to 0.45 μm, and the wavelength-independent complex refractive index m, ranges from 1.35 to 1.55 for the real part $mrr$ and from 0.005 to 0.03 for imaginary part $mri$."

We also revised **Figure 11** caption accordingly: "The reproduction of the measured PLDR and LR values, considering Chebyshev particles of the second degree ($T_2$). Purple circles correspond to measurements performed on 22 August 2017, at Leipzig, Germany, while purple lines correspond to the measurement uncertainties. Blue markers denote to simulations performed with the T-matrix code, assuming Chebyshev particles of the second degree ($T_2$). Each blue triangle corresponds to a different solution found to reproduce the measurements within their uncertainties, as given in Table 4. For these solutions, the deformation parameter $u$ ranges from -0.25 to 0.15, the mean geometric radius $r_g$ ranges from 0.2 to 0.5 μm, and the wavelength-independent complex refractive index $m$, ranges from 1.4 to 1.8 for real part $mrr$ and from 0.015 to 0.055 for the imaginary part $mri$ ."

**Figure 1 to 9** captions in the manuscript Supplement were also reviewed accordingly.

We finally added the sentence: "this is the solution highlighted in blue in Table 3" at the end of the caption of **Figure 10** and "this is the solution highlighted in blue in Table 4" at the end of the caption on **Figure 12** to be more clear that this is the solution minimizing the cost function.

[revised manuscript text omitted]